# Genome-wide CRISPR screens of oral squamous cell carcinoma reveal fitness genes in the Hippo pathway

Annie Wai Yeeng Chai[1], Pei San Yee[1†], Stacey Price[2†], Shi Mun Yee[1], Hui Mei Lee[1], Vivian KH Tiong[1], Emanuel Gonçalves[2], Fiona M Behan[2], Jessica Bateson[2], James Gilbert[2], Aik Choon Tan[3], Ultan McDermott[4], Mathew J Garnett[2], Sok Ching Cheong[1,5]*

[1]Head and Neck Cancer Research Team, Cancer Research Malaysia, Head and Neck Cancer Research Team, Subang Jaya, Selangor, Malaysia; [2]Wellcome Sanger Institute, Wellcome Genome Campus, Cambridge, United Kingdom; [3]Department of Biostatistics and Bioinformatics, Moffitt Cancer Center, Tampa, United States; [4]Oncology R&D AstraZeneca, CRUK Cambridge Institute, Cambridge, United Kingdom; [5]Department of Oral & Maxillofacial Clinical Sciences, Faculty of Dentistry, University of Malaya, Kuala Lumpur, Malaysia

**Abstract** New therapeutic targets for oral squamous cell carcinoma (OSCC) are urgently needed. We conducted genome-wide CRISPR-Cas9 screens in 21 OSCC cell lines, primarily derived from Asians, to identify genetic vulnerabilities that can be explored as therapeutic targets. We identify known and novel fitness genes and demonstrate that many previously identified OSCC-related cancer genes are non-essential and could have limited therapeutic value, while other fitness genes warrant further investigation for their potential as therapeutic targets. We validate a distinctive dependency on YAP1 and WWTR1 of the Hippo pathway, where the lost-of-fitness effect of one paralog can be compensated only in a subset of lines. We also discover that OSCCs with WWTR1 dependency signature are significantly associated with biomarkers of favorable response toward immunotherapy. In summary, we have delineated the genetic vulnerabilities of OSCC, enabling the prioritization of therapeutic targets for further exploration, including the targeting of YAP1 and WWTR1.

*For correspondence:
sokching.cheong@cancerresearch.my

†These authors contributed equally to this work

## Introduction

Head and neck squamous cell carcinoma (HNSCC) is a heterogeneous tumor arising from the mucosal surfaces lining the upper aerodigestive tract. The commonest subtype, oral squamous cell carcinoma (OSCC) is especially prevalent among Asian countries (*Bray et al., 2018*). OSCC has been associated with distinct risk habits such as betel quid chewing, tobacco smoking and alcohol consumption (*Shield et al., 2017*). The 5-year survival rate for OSCC is about 50% (*Kumar et al., 2016*) and surgery remains the mainstay of treatment. Cetuximab, an inhibitor of the epidermal growth factor receptor (EGFR), is used in combination with platinum-based chemotherapy for the treatment of advanced OSCC (*Vermorken et al., 2008*). However, the improvement in survival remains marginal (*Vermorken et al., 2008*). More recently, immune checkpoint inhibitors have been approved for the treatment of advanced and metastatic OSCC (*Cohen et al., 2019*). Although an improvement in patients' outcome is anticipated with the advancement of immunotherapy, clinical trial outcomes showed an average objective response rate of only 13–36% (*Bauml et al., 2017*; *Burtness et al., 2019*; *Ferris et al., 2016*), and the factors determining response towards checkpoint inhibitors are still largely unknown. This underscores the need to identify further therapeutic targets for OSCC.

**eLife digest** Many types of cancer now have 'targeted treatments', which specifically home in on genes cancer cells rely on for survival. But there are very few of these treatments available for the most common type of mouth cancer, oral squamous cell carcinoma, which around 350,000 people are diagnosed with each year.

Designing targeted treatments relies on detailed knowledge of the genetic makeup of the cancer cells. But, little is known about which genes drive oral squamous cell carcinoma, especially among patients living in Asia, which is where over half of yearly cases are diagnosed. One way to resolve this is to use gene editing technology to find the genes that the cancer cells need to survive.

Now, Chai et al. have used a gene editing tool known as CRISPR to examine 21 cell lines from patients diagnosed with oral squamous cell carcinoma. Most of these lines were from Asian patients, some of whom had a history of chewing betel quid which increases the risk of mouth cancer. By individually inactivating genes in these cell lines one by one, Chai et al. were able to identify 918 genes linked to the survival of the cancer cells. Some of these genes have already been associated with the spread of other types of cancer, whereas others are completely unique to oral squamous cell carcinoma. The screen also discovered that some cell lines could not survive without genes involved in a signalling pathway called Hippo, which is known to contribute to the progression of many other types of cancer.

Uncovering the genes associated with oral squamous cell carcinoma opens the way for the development of new targeted treatments. Targeted therapies already exist for some of the genes identified in this study, and it may be possible to repurpose them as a treatment for this widespread mouth cancer. But, given that different cell lines relied on different genes to survive, the next step will be to identify which genes to inactivate in each patient.

Genomic sequencing technology has enabled the delineation of the comprehensive mutational and transcriptomic landscape of cancers, including OSCC (*The Cancer Genome Atlas Network, 2015*; *Pickering et al., 2013*). However, the functional significance of most of these genetic alterations remains unclear and little is known about their value as therapeutic targets for OSCC. Identifying the genetic dependencies of OSCC will, therefore, be critical for the development of novel therapies. Genome-scale functional genetic screens allow the high-throughput identification of genes that govern cell survival (*Gerhards and Rottenberg, 2018*). Previously, such genes were identified using RNA interference (RNAi) technology (*McDonald et al., 2017*; *Tsherniak et al., 2017*). More recently, essential genes have been identified through the use of CRISPR-Cas9 technology due to its high specificity and efficiency compared to RNAi (*Gerhards and Rottenberg, 2018*). Several studies using genome-wide CRISPR-Cas9 screen have already shown promising outcome in identifying novel cancer-specific vulnerabilities that are useful drug targets (*Steinhart et al., 2017*; *Wang et al., 2017*), as well as improving the understanding of drug mechanism of action (*Barazas et al., 2018*; *Hou et al., 2017*).

The Cancer Dependency Map project (a consortium effort by the Wellcome Sanger Institute and the Broad Institute) have conducted CRISPR-Cas9 screen on a large number of cell lines including some OSCC models (*Behan et al., 2019*; *Meyers et al., 2017*). However, there is a lack of representation of Asians OSCC, such as those associated with betel quid chewing habit, one of the major risk factors of OSCC in many Asian countries (*Shield et al., 2017*). Further, the comparison of genomics data across different populations has revealed distinctive features in the different populations (*Chai et al., 2020*).

To identify genetic vulnerabilities in OSCC, we performed genome-wide CRISPR-Cas9 screens on 21 highly annotated OSCC cell lines, most of which are unique models derived from Asian patients (*Fadlullah et al., 2016*). Our study contributes to approximately one-third of the OSCC functional genetic screens currently available globally, expanding the representation of this heterogeneous disease (*Behan et al., 2019*; *Meyers et al., 2017*). In addition to finding known genetic vulnerabilities, we also uncover novel candidate genes essential for OSCC survival that can facilitate the development of new targeted therapies for OSCC. We validated the essentiality of Yes-associated protein 1 (*YAP1*) and WW domain-containing transcription regulator protein 1 (*WWTR1*) and revealed mutually

exclusive dependency and compensable functions of these paralogs in different subsets of OSCC models. We identified OSCC tumors with a gene expression signature similar to cell lines with validated dependencies. Among which, OSCC resembling the WWTR1-dependent cell lines showed significant enrichment of immune-related pathways and are associated with biomarkers of response towards checkpoint inhibitors. In summary, our study demonstrated the robustness of genome-wide CRISPR-Cas9 screen in identifying genetic vulnerabilities in diverse OSCC models, offering new molecular insights into this disease.

## Results

### Genome-wide CRISPR screens in 21 OSCC cell lines

In order to identify genetic vulnerabilities of OSCC, particularly those that are more relevant to tumors of Asian origin, we conducted genome-wide CRISPR-Cas9 knockout screens (*Figure 1A* and *Figure 1—figure supplement 1A*). We screened a unique set of 14 well-characterized OSCC cell lines termed the ORL-series [ORL-48, –115, –136, –150, –153, –156, –166, –174, –188, –195, –204, –207, -214, –215]. These were established from the tumors of Malaysian OSCC patients (*Fadlullah et al., 2016*) and are comprehensively annotated with whole-exome sequencing (WES) and RNA sequencing data. In addition, we screened a further seven OSCC cell lines [BICR10, Ho-1-u-1, HSC-2, HSC-4, PE/CA-PJ15, SAS and SCC-9] sourced from commercial cell line repositories. Demographic details of the patients from whom the 21 OSCC cell lines were derived are shown in *Figure 1A*. The presence of mutations and copy number alterations in the top five significantly mutated genes are indicated. Overall, we find that our selection of cell lines represents the diversity of mutated driver genes observed in OSCC.

### Identification of core and context-specific fitness genes

Fitness genes were identified after an unsupervised computational correction with CRISPRcleanR (*Behan et al., 2019*; *Iorio et al., 2018*), followed by mean-variance modeling and systematic ranking of significantly depleted genes using MAGeCK (*Li et al., 2014*; *Figure 1—figure supplement 1B*). At a false discovery rate (FDR) of 5%, the number of significantly depleted genes ranged from 525 genes in ORL-156 to 1399 genes in ORL-215 (*Figure 1A* and *Supplementary file 1*).

As we aimed to identify genetic vulnerabilities of OSCC that can be safely targeted therapeutically, we filtered the significantly depleted genes to exclude previously defined core fitness genes (*Behan et al., 2019*; *Hart et al., 2014*; *Hart et al., 2017*; *Meyers et al., 2017*), that were found to be essential across the many cell lines from different lineages, and are likely toxic to the cells when targeted. In general, more than 80% of all the MAGeCK hits were found to be core fitness genes and were filtered out (*Figure 1B* and *Supplementary file 1*). Overall, among the 18,010 genes screened, 2539 (14%) were found to be significant MAGeCK hits in at least one cell line and following the removal of core fitness genes (*Supplementary file 2*), 918 context-specific fitness genes were shortlisted for further prioritization (*Figure 1B*). About 40% (366 genes) were uniquely essential in a single cell line, while the remaining 60% (552 genes) were essential in at least two cell lines, hence are recurrent context-specific essentialities (*Figure 1C*).

### Capturing of known HNSCC cancer genes and pathways

Pathway enrichment analyses on the 918 context-specific fitness genes was conducted on each individual cell line (*Figure 2—figure supplement 1A*). Consistent with the highly heterogeneous nature of OSCC, diverse pathways were enriched across these cell lines. Pathways enriched among the 918 genes revealed several cancer-related pathways, potentially comprising important cancer-specific targets (*Figure 2—figure supplement 1B* and *Supplementary file 3A-B*). Whilst pathways such as ubiquitin-mediated proteolysis and cellular senescence are common across all cell lines, pathways such as NF-kappa B and MAPK signaling pathways are selectively enriched only in subsets of cell lines.

Next, we sought to determine which components of the common oncogenic pathways altered in HNSCC (*The Cancer Genome Atlas Network, 2015*) were required for cancer cell fitness and annotated the genes with the frequency of dependency (*Figure 2A*). Notably, most of the fitness genes are either existing drug targets or deemed clinically actionable. For example, drugs targeting

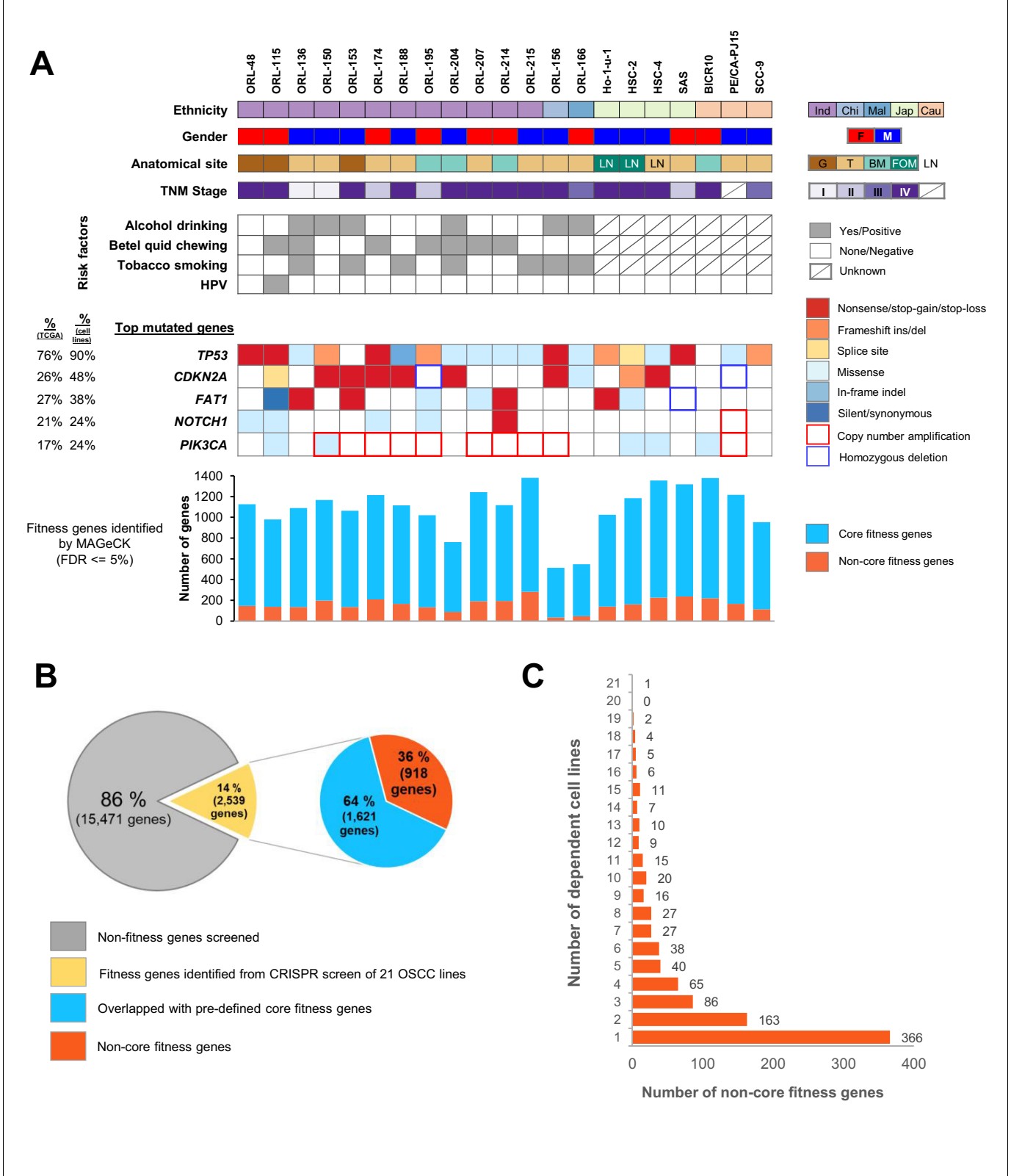

**Figure 1.** Genome-wide CRISPR-Cas9 screen on 21 OSCC cell lines. (**A**) Demographic details and genomic profile of the patients of which the 21 OSCC cell lines were derived from, with bar charts depicting the number of essential genes identified by MAGeCK. The presence of mutations/copy number alterations in the top five mutated genes in OSCC is shown. Numbers in first column indicated frequency of mutations (%) among OSCC tumors from TCGA (*The Cancer Genome Atlas Network, 2015*) while second column indicated frequency of mutation (%) among 21 OSCC cell lines. *Figure 1 continued on next page*

*Figure 1 continued*

Bar charts in the lower panel shows the number of significant fitness genes (those with MAGeCK FDR less than or equal to 5%), with the orange bars representing the number of non-core fitness genes. *Abbreviations: Ind – Indian; Chi – Chinese, Mal – Malay; Jap – Japanese, Cau – Caucasian; F – Female; M – Male; G – Gingiva; T – Tongue; BM – Buccal Mucosa; FOM – Floor of Mouth; LN – derived from lymph node metastasis.* (B) Pie charts showing the proportion of fitness genes among the 18,010 genes screened. 918 non-core fitness genes were shortlisted after filtering out the core fitness genes. (C) Bar chart depicting the number of non-core fitness genes that are found in 1 to 21 dependent cell lines.

The online version of this article includes the following source data and figure supplement(s) for figure 1:

**Source data 1.** Analysis result from the genome-wide CRISPR-Cas9 screens.
**Figure supplement 1.** Genome-wide CRISPR-Cas9 screen.

---

*PIK3CA* and *CDK6* are already in clinical trials for HNSCC treatment [NCT01816984, NCT02537223, NCT03356223, NCT03356587]. We also examined the dependency profile of 44 cancer genes with driver mutations known to be associated with HNSCC (*Bailey et al., 2018*; *Martincorena et al., 2018*; *Figure 2B*) and found that more than half of these cancer genes were dispensable for OSCC survival. Oncogene addiction has been the promising source of finding the Achilles heel for successful molecular targeted therapy (*Weinstein and Joe, 2008*). Based on the WES data of the 21 OSCC cell lines, we found 43 genes with driver mutations in at least one cell line and plotted the CRISPR score (measure of sgRNA depletion in the CRISPR screen) to examine if there is any differential dependency associated with the mutations (*Figure 2C* and *Supplementary file 4*). Dependencies on mutated *PIK3CA* were observed in four OSCC cell lines with a hotspot mutation in E545K (BICR10), Q546R (ORL-150) and H1047R (HSC-2), and to a lesser extent, in ORL-115 with H1047L mutation. Intriguingly, HSC-4, which harbors the same E545K mutation as BICR10, did not show any dependency on the mutated *PIK3CA*, this is consistent with findings from Project Score (*Behan et al., 2019*). A splice site driver mutation in *PTEN* co-occurred in HSC-4 and may have counteracted the oncogene addiction effect on the mutated *PIK3CA*, as suggested previously in breast cancer (*Lazaridis et al., 2019*). Dependency on NFE2L2 was observed in Ho-1-u-1 and BICR10 however the mechanism of activating this oxidative pathway differed between these two cell lines. Oncogene addiction is observed for Ho-1-u-1 with a hotspot mutation (E82D) in *NFE2L2*, that has been shown to enhance its transcriptional activity and promoting cell proliferation (*Shibata et al., 2008*). On the other hand, BICR10 harbors an inactivating mutation on *KEAP1* (R320Q) a negative regulator of *NFE2L2*. The R320Q mutation has been reported to stabilize NRF2 (encoded by *NFE2L2*) and enhances cell fitness as reported previously in lung cancer (*Hast et al., 2014*). Finally, the only cell line that shows dependency on HRAS is ORL-214 which carries a mutation in *HRAS* (G12C). Another gene, encoding ZFP36L1 with truncating mutation at S324 in ORL-48 also showed preferential dependency, suggesting that the effect of this mutation should be studied further.

## Identification of unique dependencies in betel-quid-associated OSCC

Studies on the genomic landscape of Asian and Caucasian OSCC have revealed distinct molecular differences, suggesting that some population-specific risk habits might have contributed to these differences (*The Cancer Genome Atlas Network, 2015*; *Chai et al., 2020*; *Hsieh et al., 2001*; *India Project Team of the International Cancer Genome Consortium, 2013*; *Zanaruddin et al., 2013*). Betel quid chewing is frequently associated with OSCC in Asia (*Guha et al., 2014*; *India Project Team of the International Cancer Genome Consortium, 2013*; *Shah et al., 2012*). In this study, several Asian-derived OSCC models that were associated with betel-quid chewing were included, and we had the opportunity to determine if there are differences in genetic dependencies between these OSCC (ORL-115, ORL-136, ORL-174, ORL-195, ORL-204, ORL-207, ORL-214) (n = 7), with those that are not associated with betel quid chewing (n = 14) (*Figure 2D*). Of the 110 fitness genes uniquely seen in betel-quid-associated OSCC, the NF-kB signaling pathway stands out as one of the significantly enriched pathway. The fitness genes from this pathway that are unique to the betel quid-associated OSCC include *NFKB2, TNFAIP3, CSNK2A1,* and *TRIM25*. When cross checking with the DepMap and Project Score data, three out of four (75%) of these genes (*NFKB2, TNFAIP3,* and *TRIM25)* were not found among the screened OSCC models, which were mostly derived from Caucasians, or from Asians not known to chew betel quid. Interestingly, our findings are coherent with previous studies from Taiwan and India, where betel quid chewing is common,

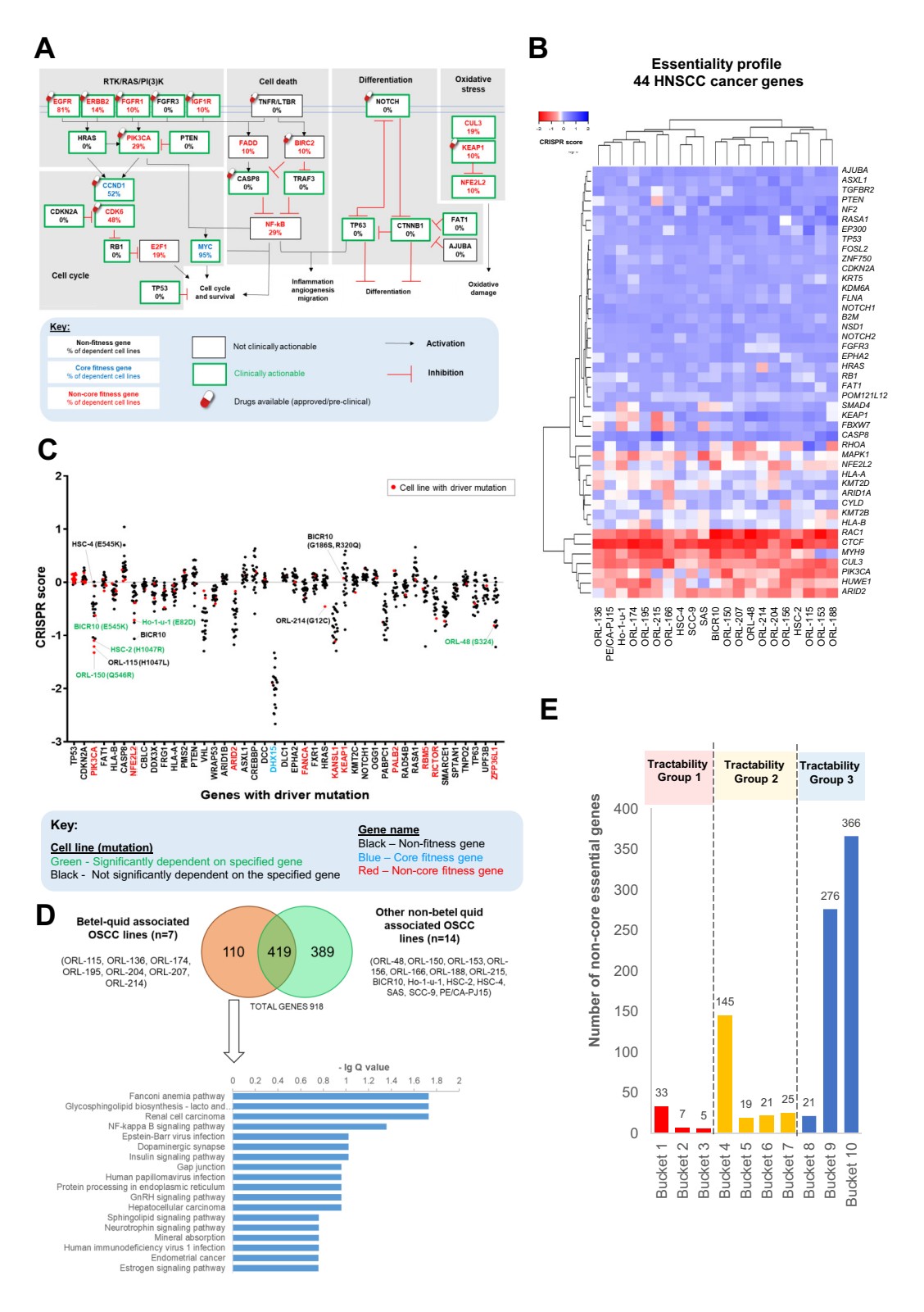

**Figure 2.** Identification and analysis of targetable genes and pathways in OSCC. (**A**) Common oncogenic pathways altered among HNSCC samples of TCGA were annotated with frequency of dependency. Non-core fitness genes are indicated in red and the percentage of OSCC cell lines that were dependent on the genes are shown. The Drug Gene Interaction database (DGIdb) (http://www.dgidb.org/) was used to determine if the gene is clinically actionable while the availability of drugs were determined using Open Targets Platform (https://www.targetvalidation.org/). (**B**) Heatmap of

*Figure 2 continued on next page*

*Figure 2 continued*

gene essentiality of the 44 HNSCC cancer genes in the 21 OSCC lines. These are consensus cancer genes for HNSCC curated from *Bailey et al., 2018* and *Martincorena et al., 2018*. (C) CRISPR scores of genes with driver mutations in at least one of the 21 OSCC cell lines. Cell lines labeled in green with mutation are examples of those showing oncogene addiction on mutated genes, for example on *PIK3CA* (ORL-150, BICR10, and HSC-2) and *NFE2L2* (Ho-1-u-1). (D) Pathway enrichment analysis for fitness genes that are differentially enriched among the seven betel-quid associated OSCC. (E) Distribution of the 918 fitness genes based on their small molecule inhibitors tractability assessment. Tractability is defined as detailed in *Behan et al., 2019* where: tractability group one included targets with approved drugs (Bucket 1) or drugs in clinical/pre-clinical development (Bucket 2, 3); tractability group two included targets with evidence supporting tractability albeit no drugs are available yet; while the least tractable group three included targets that lacks evidence informing tractability.

The online version of this article includes the following source data and figure supplement(s) for figure 2:

**Source data 1.** Analysis results of targetable genes and pathways in OSCC.

**Figure supplement 1.** Pathway level enrichment analysis and tractability assessment of fitness genes.

which have demonstrated that extract from the areca nut of betel quid can directly activate the NF-kB signaling pathway, favoring OSCC cells survival (*Chiang et al., 2008*; *Islam et al., 2019*; *Lin et al., 2005*).

## Tractability of the identified fitness genes

Given that many of the reported HNSCC-related cancer genes do not appear to be fitness genes, we sought to determine which of the 918 genes could potentially be tractable using previously defined frameworks (*Behan et al., 2019*; *Brown et al., 2018*; *Figure 2E* and *Supplementary file 5*). From the 918 genes, 45 genes fall into the tractability group 1, some examples of genes in this group include *EGFR*, *PIK3CA*, *CDK4*, and *CDK6*, where anticancer drugs targeting these genes are already approved or clinical trials for the treatment of HNSCC are on-going, demonstrating the robustness of our results. When classified based on protein function using PANTHER (*Mi et al., 2013*), most of those in tractable group 1, are transferase (kinases) (32%) and oxidoreductase (30%), including genes like *CDK4*, *CDK6* and *PIK3CA*; and several genes in the family of NADH:ubiquinone oxidoreductase such as *NDUFB9* and *NDUFC2* (*Figure 2—figure supplement 1C*). Interestingly, emerging oncology and non-oncology drugs such as the HDAC inhibitors (*Yoon and Eom, 2016*) and miglustat, an approved drug for Gaucher's disease (*Barth et al., 2013*), were amongst potential drug repurposing candidates that target fitness genes in tractability group 1 (targeting *HDAC2* and *UGCG* respectively). The only transcription factor that falls within tractability group one is the *ESR2*, with several antagonists available owing to its well-studied ligands and structure (*Ho, 2004*; *Figure 2—figure supplement 1C*). A further 210 genes (23%) belong to tractability group 2, which harbors novel targets that have evidence supporting their tractability. Albeit no drugs are currently in clinical trials, these may hold potential for future drug development. For example, several companies are developing drugs that could target YAP1, SLC2A1 and PTPN11 which are in tractability group 2. However, about 70% of the 918 genes belong to tractability group 3 (least tractable or lacking evidence) where significant efforts in understanding their structure and function would be necessary to evaluate their tractability. Consistent with previous reports (*Behan et al., 2019*), the least tractable group comprises mainly of nucleic acid-binding proteins and transcription factors, such as the Kruppel-like factors (KLF) gene families (*KLF4* and *KLF5*) and zinc finger proteins (*ZNF148* and *ZNF236*) (*Figure 2—figure supplement 1C*).

## Fitness genes in copy number amplified regions

HNSCC belongs to the 'C class' tumor, where the landscape of genomic alterations is dominated by copy number alterations including recurrent chromosomal gains and losses (*Ciriello et al., 2013*). The most frequently reported copy number gains occurr in chromosomes 3q, 5 p, 7 p, 8q, and 11q (*The Cancer Genome Atlas Network, 2015*; *Salahshourifar et al., 2014*). To identify putative oncogenes that are essential for OSCC within these amplified regions, we evaluated the number of candidate fitness genes before and after CRISPRcleanR correction for copy number bias (*Figure 3—figure supplement 1A*). After correction, no enrichment/bias was found in the frequently amplified chromosome, demonstrating effective correction of copy number bias (*Figure 3—figure supplement 1B*). KEGG pathway analysis of the 152 genes from the five amplified regions showed enrichment of several oncogenic signaling pathways such as the small cell lung cancer, Hippo signaling

pathway and ErbB signaling pathway (*Figure 3—figure supplement 2A*). We focused our analysis on the Hippo signaling pathway (*Figure 3—figure supplement 2B*), which has recently been implicated for major oncogenesis role in squamous cell carcinoma, including OSCC (*Campbell et al., 2018*; *Ge et al., 2011*; *Hiemer et al., 2015*; *Wang et al., 2018*). Furthermore, *YAP1* or *WWTR1* amplifications occur in approximately 19% of HNSCC (*YAP1*–5.5%, *WWTR1*–14.3%), which puts it among the top five cancers with the highest amplification of these genes amongst 33 cancer types (*Wang et al., 2018*).

## Differential dependency pattern on YAP1 and WWTR1

YAP1 and WWTR1 (also known as TAZ) are transcription co-activators, which are the major effectors of the Hippo pathway (*Guo and Teng, 2015*; *Wang et al., 2018*). YAP1 and WWTR1 are paralogs with ~46–60% similarity in their amino acid sequence (*Guo and Teng, 2015*). They were shown to have both overlapping and distinct roles in different contexts (*Guo and Teng, 2015*; *Plouffe et al., 2018*). In OSCC, overexpression of YAP1 and WWTR1 has been shown to increase proliferation, survival and migration, mainly via interaction with the transcriptional enhanced associate domain (TEAD) transcription factors (*Hiemer et al., 2015*).

Interestingly, across the 21 OSCC cell lines, there is a subset of lines that show significant dependency on only one of the paralogs, while another subset of lines does not exhibit significant dependency on either *YAP1* or *WWTR1* (*Figure 3A*). With the exception of two cell lines, ORL-153 and ORL-215, YAP1-dependent lines and WWTR1-dependent lines are mutually exclusive. There were seven lines that were highly dependent on YAP1 (ORL-48, ORL-136, ORL-156, ORL-204, ORL-207, SAS, SCC-9), four highly dependent on WWTR1 (ORL-174, ORL-188, ORL-214, PE/CA-PJ15), while the other eight were not dependent on either (*Figure 3A*). Notably, *WWTR1* gene locus (3q25) is at close proximity to the locus of *PIK3CA*, *SOX2* and *TP63* at 3q26-28, whereby their focal amplification is frequently reported in HNSCC (*The Cancer Genome Atlas Network, 2015*; *Figure 3—figure supplement 2C*). The majority of the WWTR1-dependent cell lines have copy number amplification for these genes on 3q25-28 (*Figure 3A*), but only WWTR1 CRISPR scores are significantly different between those with and without copy number amplification (p<0.001) (*Figure 3—figure supplement 2D*). Only two of nine cell lines with PIK3CA copy number amplification are dependent on PIK3CA itself. This suggests that copy number amplification of WWTR1 may constitute to a functional oncogenic role of WWTR1 in OSCC, instead of being a passenger gene that is co-amplified with the canonical HNSCC oncogene, *PIK3CA*. Notably, we also observed an enrichment of *PIK3CA* mutations (p=0.0003) among cell lines that are neither dependent on YAP1 or WWTR1, whereby five out of six such lines have PIK3CA hotspot mutations (BICR10, HSC-2, HSC-4, ORL-115, and ORL-150) (*Figure 3A*).

Intriguingly, mutually exclusive copy number gains of chromosome 3q and 11q22 (where YAP1 is mapped to) have been reported in squamous cell carcinoma (*Campbell et al., 2018*) and consistently, YAP1 and WWTR1 amplification were also found to be mutually exclusive in HNSCC (*Wang et al., 2018*). In our study, despite having cell lines dependent on YAP1, none of the 21 OSCC cell lines shows copy number amplification of YAP1 or neighbouring genes on the chromosome 11q22. This suggests that other non-genomic mechanisms could likely be in place to activate YAP1.

To investigate if expression of *YAP1* and *WWTR1* is associated with the dependency, we examined baseline mRNA and protein expression of YAP1 and WWTR1 in representative OSCC lines (*Figure 3—figure supplement 2E–F*). Among the YAP1-dependent lines (ORL-48, ORL-204), overexpression of YAP1 mRNA and protein levels were observed. Further, these lines have low protein expression of WWTR1. The mRNA and protein expression of WWTR1 are relatively higher than YAP1 among the WWTR1-dependent cell lines (ORL-214 and PE/CA-PJ15), and those that were not affected when either YAP1 or WWTR1 is knocked-out ('non-dependent'). To determine if the association between dependency and gene expression is exclusive to OSCC, we computed the differential dependency score of *YAP1* and *WWTR1* and their gene expression for 273 cancer cell lines from Project Score (*Behan et al., 2019*; *Figure 3—figure supplement 3A*). Generally, cell lines that are dependent on YAP1 have higher YAP1 expression, and similarly, WWTR1-dependent lines have higher expression of WWTR1 compared to YAP1. WWTR1 gene expression showed significant negative correlation with its dependency in these 273 cancer cell lines (Pearson R = −0.570, p-value=3e-25) and showed non-significant negative correlation among our 21 OSCC lines screened (Pearson

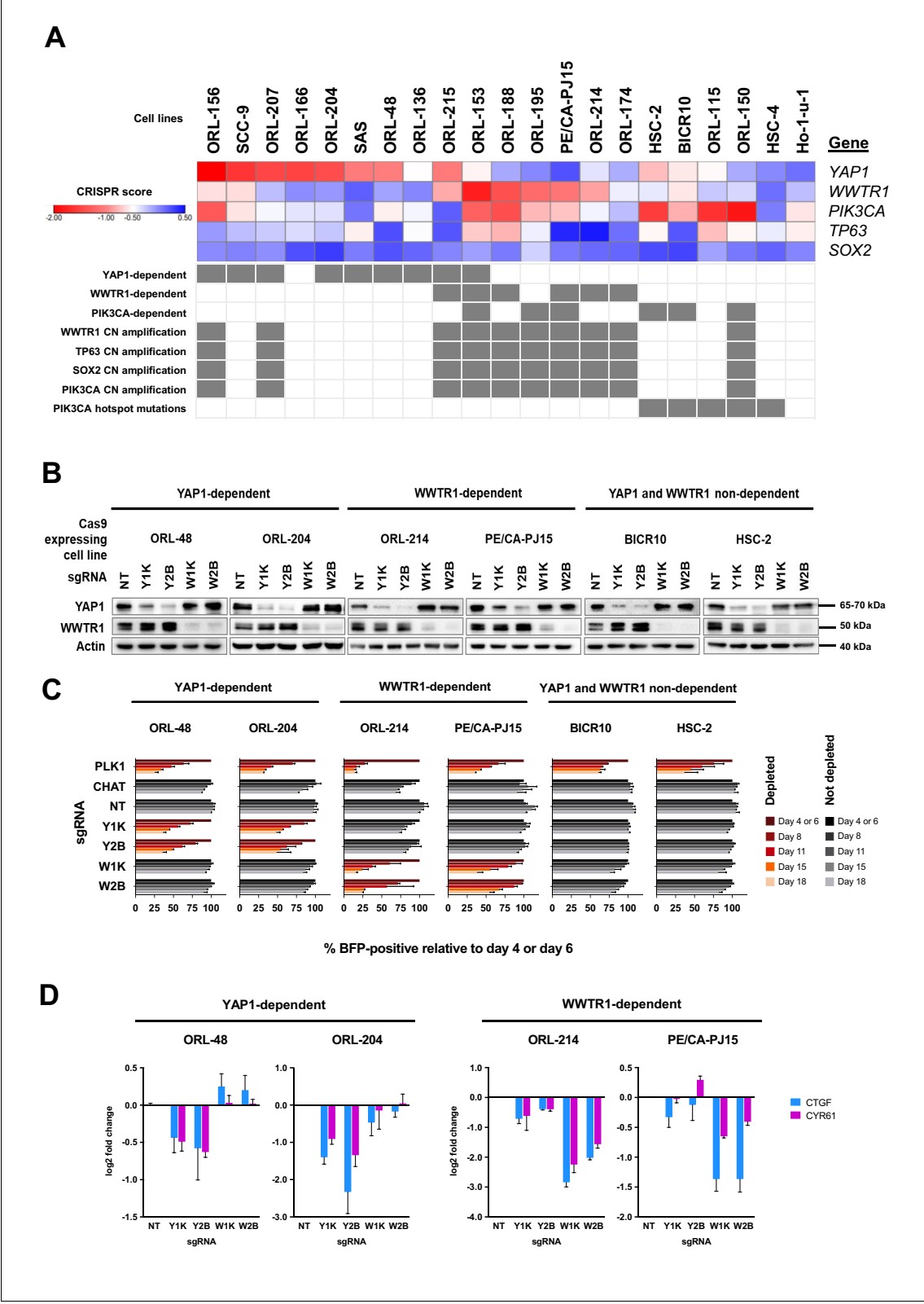

**Figure 3.** YAP1 and WWTR1 of the Hippo pathway are selectively essential in a distinct subset of OSCC cell lines. (**A**) Essentiality profile (depicted with CRISPR scores heatmap) of YAP1, WWTR1, PIK3CA, TP63 and SOX2 across 21 OSCC cell lines derived from the CRISPR/Cas9 screen. Dependency on these genes were depicted as grey box in the bottom panel, according to the MAGeCK definition of significant depletion at FDR ≤ 0.05. No cell lines were dependent on TP63 or SOX2. The degree of essentiality differs across the lines. Some subsets of the cell lines are only dependent on either YAP1

*Figure 3 continued on next page*

*Figure 3 continued*

or WWTR1, while neither gene appears to be essential in another subset of cell lines. *PIK3CA, TP63* and *SOX2* are genes implicated in HNSCC carcinogenesis that are often co-amplified with WWTR1, located on chromosome 3q25-28. All WWTR1-dependent cell lines had copy number amplification on these genes while all PIK3CA mutated cell lines are not dependent on either YAP1 or WWTR1. (B) Western blot images showing the protein level of YAP1 and WWTR1 on day 4 upon transducing the Cas-9 expressing cell lines with lentivirus carrying gene-specific sgRNA. Two sgRNAs were used per target gene. (C) Co-competition assay was used to validate the essentiality of YAP1 and WWTR1 on the selected cell lines. The growth of the BFP-positive transduced population was compared to the non-transduced population throughout the 18 days assay. The percentage of BFP-positive cells obtained at different time points were normalized to the day 4 readings for respective sgRNA (except ORL-204 which had time points normalized to the day 6 readings for respective sgRNA). PLK1 is a core essential gene included as a positive control. Negative controls include CHAT which is a non-essential gene across the panel of cell lines, and NT serves as a non-targeting control. Data are shown as mean ± SD (n = 2 biological repeats). (D) qPCR results show suppression of downstream targets of YAP1 and WWTR1 only when the respective fitness gene is being knocked-out. Down-regulation of CTGF and CYR61 gene expression was observed when YAP1 is knocked-out in the YAP1-dependent cell lines (ORL-48 and ORL-204). In the WWTR1-dependent cell lines (ORL-214, PE/CA-PJ15), CTGF and CYR61 expression is only suppressed when WWTR1 is knocked-out. Data are shown as mean ± SD (n = 2 independent experiments with technical triplicates).

The online version of this article includes the following source data and figure supplement(s) for figure 3:

**Source data 1.** All raw data related to *Figure 3* and its figure supplements on analysis result of YAP1 and WWTR1 as fitness genes for OSCC.
**Figure supplement 1.** Fitness genes and copy number amplification.
**Figure supplement 2.** YAP1 and WWTR1 from the Hippo signaling pathway are fitness genes for OSCC.
**Figure supplement 3.** Correlation of gene essentiality and gene expression for YAP1/WWTR1.
**Figure supplement 4.** Clonogenic assay of representative lines and validation of YAP1 dependency in SAS.

R = −0.354, p-value=0.116) (*Figure 3—figure supplement 3B–C*). In other words, the higher dependency on *WWTR1* gene is associated with higher *WWTR1* gene expression. Interestingly, this observation was also seen in other cancer types, including non-small cell lung carcinoma, squamous cell lung carcinoma, glioblastoma, breast carcinoma, and lung adenocarcinoma (*Figure 3—figure supplement 3D*), based on the Project Score data. Among these cancers, non-small cell lung carcinoma showed the highest percentage of WWTR1-dependency (30%), with the strongest correlation (Pearson's R = −0.934, p-value=0.0021).

To validate the differential dependency for YAP1 and WWTR1, we performed single-gene knock-out using two sgRNAs per gene and investigated the growth inhibition effect of gene knockout using co-competition assay, as previously reported (*Behan et al., 2019*). Two cell lines each from the YAP1-dependent (ORL-48 and ORL-204), WWTR1-dependent (ORL-214 and PE/CA-PJ15) and YAP1/WWTR1 non- dependent (BICR10, HSC-2) groups were used. The efficacy of protein knockout using individual sgRNAs was assayed with western blotting (*Figure 3B*).

The results obtained from the co-competition assay corroborated our CRISPR screen data (*Figure 3C*). More than half of the transduced cell population was depleted following YAP1-knock-out in ORL-48 and ORL-204 cells, but not when WWTR1 was knocked-out. Likewise, the growth inhibition in ORL-214 and PE/CA-PJ15 was only seen following WWTR1-knockout, but not upon YAP1-knockout. On the other hand, the fraction of transduced cells in BICR10 and HSC-2 did not show any prominent changes upon knockout of either YAP1 or WWTR1. These experiments were validated using clonogenicity assays (*Figure 3—figure supplement 4A*). Together, these results support the differential dependency pattern on *YAP1* and *WWTR1*. Besides, another YAP1-dependent line, SAS has recently been reported to harbor a fusion protein of YAP1 and MAML2 (*Picco et al., 2019*). Our co-competition assay also confirms the dependency on YAP1 in these cells and the differential depletion of sgRNA targeting early and late exons of YAP1 suggested that this oncogenic fusion protein provided a survival advantage (*Figure 3—figure supplement 4B–E*). In OSCC, *CTGF*, and *CYR61* are two canonical transcriptional targets of YAP1 and WWTR1 (*Hiemer et al., 2015*). Consistent with their known pro-survival properties, more substantial reduction of *CTGF* and *CYR61* gene expression were seen only when the respective upstream fitness genes were knocked-out, as measured by qPCR (*Figure 3D* and *Figure 3—figure supplement 4F*).

Since YAP1/WWTR1 were both known to regulate proliferation and apoptosis, we next investigated whether the depletion of YAP1 and WWTR1 affects cell proliferation or apoptosis in the selected OSCC cells. Consistently, YAP1 depletion in ORL-48 resulted in significant reduction in viable cells, to a level comparable to the depletion of PLK1 (*Figure 4A*), and this was reflected in the increase in apoptotic cells (*Figure 4B*). Further, WWTR1 depletion in ORL-214 showed significantly

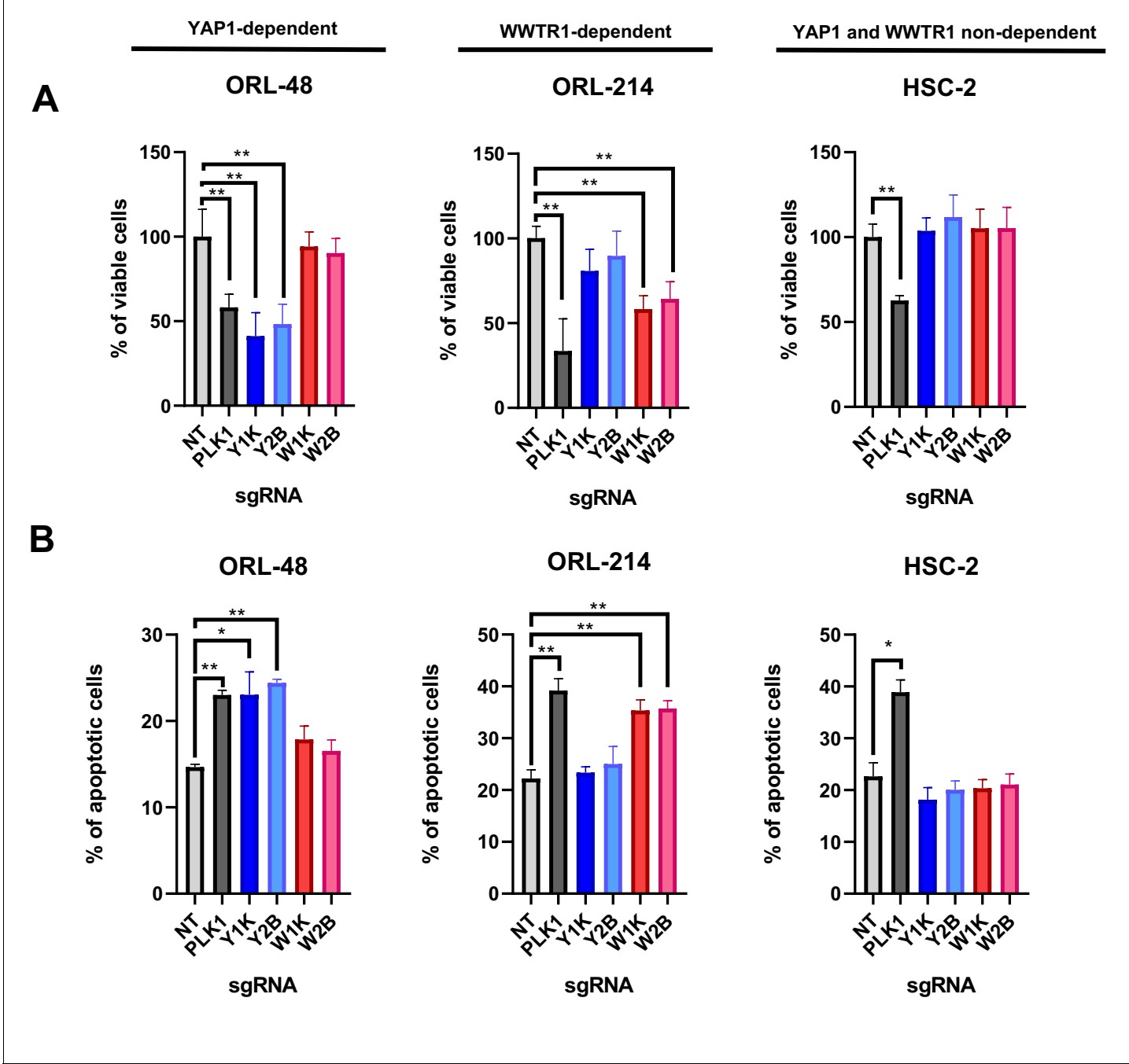

**Figure 4.** YAP1/WWTR1 knockout impairs proliferation and induces apoptosis in dependent cells. (**A**) Knocking out YAP1 or WWTR1 significantly inhibited cell proliferation in the respective dependent cell lines. Cell viability of HSC-2 (a YAP1- and WWTR1-non-dependent cell line) was not affected by depletion of either YAP1 or WWTR1. Experiments are repeated twice in technical triplicates. (n = 3 technical repeats for two biological repeats). Data shown as mean ± SD. *p<0.01; **p<0.001. (**B**) Apoptosis assay revealed that YAP1 depletion in YAP1-dependent ORL-48 resulted in significant increase in apoptotic cells while WWTR1 depletion in WWTR1-dependent ORL-214 resulted in an increase in apoptotic cells. No significant changes in apoptotic cells were observed in HSC-2 when either YAP1 or WWTR1 was depleted. Experiments are repeated twice in technical triplicates. Data of one experiment are shown as mean ± SD (n = 3 technical repeats). *p<0.01; **p<0.001.

The online version of this article includes the following source data for figure 4:

**Source data 1.** Data for cell viability and apoptosis assays.

lower percentage of viable cells when compared with the control, as well as significant increase in apoptotic cells (*Figure 4A–B*). Overall, these results confirm the dependency on either YAP1 or WWTR1 for survival, whereby depletion of the respective fitness genes resulted in increased apoptosis. By contrast, depletion of either YAP1 or WWTR1 did not affect the survival of HSC-2, as confirmed by the apoptotic assay (*Figure 4A–B*).

## YAP1 and WWTR1 shows compensatory roles in OSCC cell lines

The lack of dependency on either YAP1 or WWTR1 in the non-dependent lines is intriguing, given the importance of these genes in most OSCC cancer cells. As YAP1 and WWTR1 share high structural homology and have common downstream targets, we hypothesized that YAP1 and WWTR1 could provide compensatory functions to maintain the survival of BICR10 and HSC-2 cell lines when either one of the genes was knocked-out.

To confirm our hypothesis, we knocked-out both YAP1 and WWTR1 simultaneously by co-transducing the cell lines with lentivirus carrying blue fluorescence protein (BFP)-tagged YAP1 sgRNA and mCherry-tagged WWTR1 sgRNA (*Figure 5A*). The co-competition assays showed that the population of BICR10 and HSC-2 with the double knockout of both YAP1 and WWTR1 depleted drastically (*Figure 5B*) compared to when each gene was knocked-out individually (*Figure 3C*). This suggests that in this YAP1/WWTR1 compensable subset of cell lines, the paralogs can compensate for the function of one another to activate the downstream mechanisms required to maintain cell fitness. This was substantiated by quantitative-PCR (qPCR) of downstream targets CTGF and CYR61 where substantial down-regulation in double knockout cells were observed compared to when each gene is knocked-out individually (*Figure 5C*).

## YAP1/WWTR1-dependency associated gene signatures in OSCC

Next, we sought to determine whether the differential dependency on YAP1 and WWTR1 is also relevant in OSCC tumors. We first derived the gene expression signatures representing the three groups using differentially expressed gene (DEG) analysis based on the OSCC cell lines with validated dependency (*Figure 6—figure supplement 1A–B* and *Supplementary file 6*). Using the 'YAP1 dependency signature score', 'WWTR1 dependency signature score' and 'Compensable signature score' [see materials and methods], OSCC cell lines were clustered into three broad groups based on their dependency on YAP1, or WWTR1 (*Figure 6A*). Using the same algorithm, we then computed the dependency signature score for each of the 315 OSCC tumors from the TCGA HNSCC cohort. From the heatmap and clustering analysis based on their dependency signatures, the three groups were also observed among the OSCC tumors (*Figure 6B*). To define representative 'core' samples, we found 41 OSCC tumors (13%) with high YAP1 dependency signature score (>0.5); 30 OSCC (9.5%) with high WWTR1 dependency signature score (>0.5) and 34 OSCC (11%) with high Compensable signature score (>0.5). Using these core OSCC samples and cell lines with validated dependency, we then used GSEA to identify hallmark pathways that are enriched in each of these groups (*Figure 6C* and *Supplementary file 7*). YAP1-dependent cell lines and tumors showed enrichment in hallmarks related to cell cycle, such as the E2F targets, G2M checkpoint, MYC targets, and DNA repair pathways (*Figure 6C*). This is consistent with previous reports that have demonstrated that the transcription factors E2F and MYC are critical downstream regulators of YAP/TEAD-mediated activation of cell cycle genes (*Kapoor et al., 2014*; *Pattschull et al., 2019*). While OSCC with high WWTR1 dependency signature score showed high expression of genes in several hallmarks related to immunity, such as the interferon responses, inflammatory responses and the complement pathway. This association is aligned with the recent findings that WWTR1 may play a role in immunity by upregulating PD-L1 expression (*Janse van Rensburg et al., 2018*). On the other hand, cell lines and tumors with high Compensable signature score showed enrichment in several metabolism-related hallmarks, such as fatty acid metabolism and xenobiotic metabolism. Notably, all three YAP1/WWTR1 compensable cell lines (BICR10, HSC-2 and HSC-4) harbor *PIK3CA* mutation, and that alterations in the PI3K signaling pathway have been linked to multiple metabolic dysregulations in cancer (*Hao et al., 2016*).

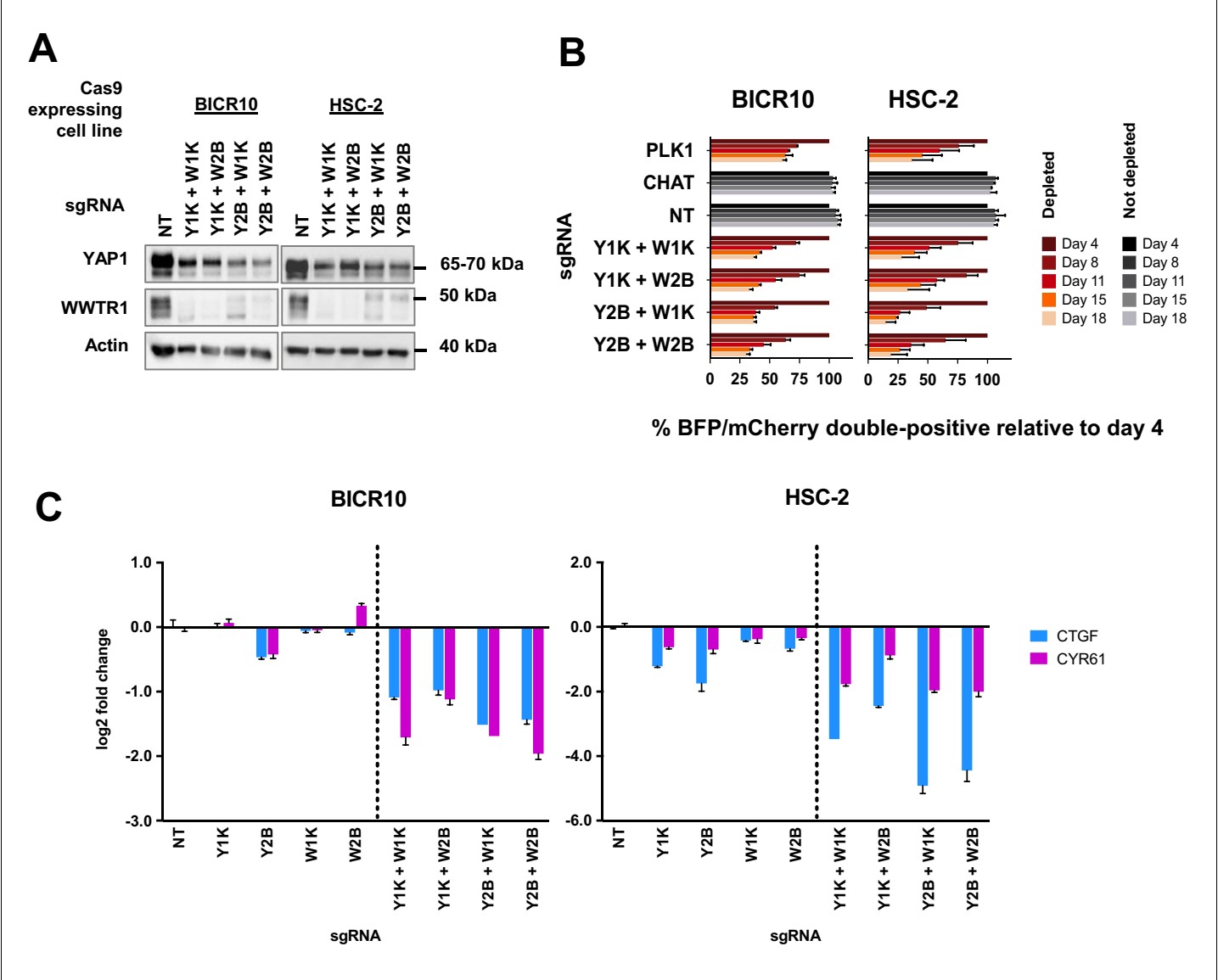

**Figure 5.** Simultaneous depletion of YAP1 and WWTR1 on BICR10 and HSC-2 inhibited cell growth, suggesting compensatory role of YAP1 and WWTR1. (**A**) Western blot showing the protein level of YAP1 and WWTR1 in BICR10 and HSC-2 Cas9 cell lines upon different combinations of YAP1 and WWTR1 sgRNA co-transduction. sgRNAs that targeted YAP1 were tagged with BFP marker and WWTR1 sgRNAs were tagged with mCherry marker respectively, they were simultaneously transduced into the Cas9-expressing cell lines to achieve double knockout of YAP1 and WWTR1. (**B**) YAP1 and WWTR1 double knockout cells show inhibited growth in co-competitive assay. Simultaneous knockout of YAP1 and WWTR1 in Cas9-expressing BICR10 and HSC-2 resulted in the depletion of sgRNAs-transduced populations, which was not observed upon the depletion of either gene alone. Results shown were normalized to day 4. Data are shown as mean ± SD (n = 2 independent experiments with technical triplicates). (**C**) qPCR results revealed that strong suppression of *CTGF* and *CYR61* gene expression in BICR10 and HSC-2 is only seen when YAP1 and WWTR1 are simultaneously knocked-out. qPCR was performed in technical triplicates. Data are shown as mean ± SD (n = 3 technical repeats). The online version of this article includes the following source data for figure 5:

**Source data 1.** Data for co-competitive assay and qPCR.

## OSCC with WWTR1 dependency signature and immune biomarkers

Among the plethora of diverse functions for YAP1 and WWTR1 evidence for their critical roles in mediating immune response have recently emerged (*Geng et al., 2017*; *Janse van Rensburg et al., 2018*; *Pan et al., 2019*). Specifically, WWTR1 but not YAP1 was shown to be essential for $T_H17$ cell differentiation (*Geng et al., 2017*), and constitutively active WWTR1 (TAZ-S89A) was shown to induce PD-L1 expression to a much greater extent than YAP1 (YAP1-S127A) (*Janse van Rensburg*

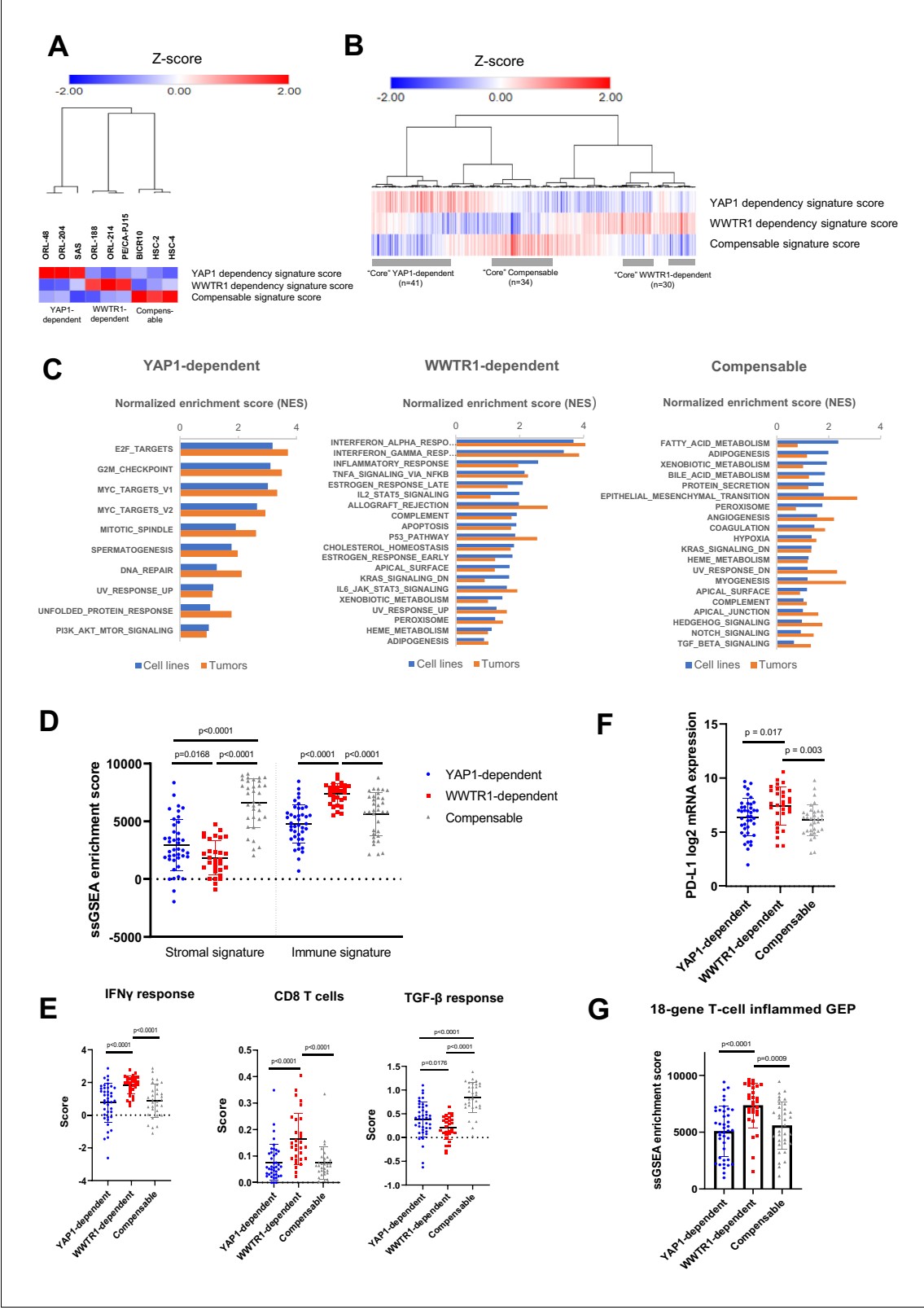

**Figure 6.** Identification of YAP1/WWTR1 dependency associated gene signatures in OSCC tumors samples. (**A**) Heatmap with hierarchical clustering using the computed signature scores showed clustering of the OSCC cell lines into three groups, based on their validated dependency. (**B**) Heatmap with hierarchical clustering of OSCC tumors using the computed signature scores. The three clusters found from cell lines were also present among the OSCC tumors. For labeling convenience, OSCC with high YAP1 dependency signature score is referred as 'YAP1-dependent'; 'WWTR1-dependent' –

*Figure 6 continued on next page*

*Figure 6 continued*

high WWTR1 dependency signature score; 'Compensable' – high compensable signature score. (C) GSEA of the OSCC cell lines and those OSCC tumors with similar gene signatures showed overlapping hallmarks enrichment, while distinct hallmarks were associated with each of the three groups. Only common hallmarks (between cell lines and tumors) with positive normalized enrichment score (NES) of >0.5 were shown. Full GSEA results can be found in *Supplementary file 7*. (D) OSCC with high WWTR1 dependency signature score showed significantly lower stromal infiltration, but higher immune infiltration compared to the other two groups. (E) By comparing the immune expression signatures from *Thorsson et al., 2018*, OSCCs with high WWTR1 dependency signature score are associated with significantly higher IFN-gamma response and cytolytic CD8 T-cells and lower TGF-beta response. (F) mRNA expression of PD-L1 (CD274) were significantly elevated among OSCC tumors with high WWTR1 dependency signature score. (G) OSCC with high WWTR1 dependency signature score showed significantly higher enrichment score for the 18-gene T-cell inflamed GEP, which is a clinically validated biomarker of response towards checkpoint blockade. For panels D-G, unpaired Welch's t-test with Welch's correction was used due to unequal sample size (n = 43, 31, 30, respectively).

The online version of this article includes the following figure supplement(s) for figure 6:

**Figure supplement 1.** Analysis workflow for computing dependency score and heatmap of gene expression for the differentially expressed genes.
**Figure supplement 2.** Comparison of lymphocytes score, immune expression signature and tumor mutational burden.
**Figure supplement 3.** Association with the immune-related gene *PD-L1* is specific to *WWTR1* but not other co-amplified genes in chromosome 3q25-28.

---

*et al., 2018*). Consistently, the results from the enrichment analysis of WWTR1-dependent OSCC were dominated by immune-related hallmarks. To investigate the association of YAP1/WWTR1 dependency with the immunity of OSCC, we mined the previously defined immune landscape of core samples from TCGA for comparison. Using the stromal and immune signature defined previously (*Yoshihara et al., 2013*), ssGSEA enrichment scores revealed that OSCC with high WWTR1 dependency signature score showed significantly lower stromal signature, but higher immune signature, when compared with OSCC of high YAP1 dependency or Compensable signature scores (*Figure 6D*). This suggests an enrichment of immune cell infiltration among OSCC with high WWTR1 dependency signature score. Next, we also assessed the other immune-related scores from previous work (*Thorsson et al., 2018*) and confirmed that these OSCC were associated with significantly enriched interferon-gamma (IFNγ) response signatures and reduced transforming growth factor-beta (TGFβ) response (*Figure 6E* and *Figure 6—figure supplement 2A–B*). Consistent with that, they also possess significantly higher cytolytic T cells (CD8) score than the other two groups (*Figure 6E*). This interesting observation led us to postulate that OSCC cancers with a gene expression signature associated with WWTR1 dependency might be more vulnerable to checkpoint blockade. To extend this hypothesis, we examined the level of several predictive biomarkers for immune checkpoint inhibitor response that had been tested and validated in the clinical setting (*Ayers et al., 2017*; *Cristescu et al., 2018*). Interestingly, the PD-L1 mRNA expression and the 18-gene T-cell inflamed gene expression profile (GEP) enrichment scores of the OSCC with high WWTR1 dependency signature score were significantly higher than the other two groups (*Figure 6F–G*). They were also associated with higher tumor mutational burden (TMB) albeit not statistically significant when compared with the other two groups (*Figure 6—figure supplement 2C*). To ensure that the association with immune signatures seen is specific to WWTR1 but not caused by other co-amplified genes in 3q25-28 (such as *PIK3CA*, *TP63*, and *SOX2*), we examined the changes in gene expression level of *PD-L1* upon depletion of *WWTR1* and other co-amplified genes. qPCR results revealed strongest suppression of *PD-L1* gene expression upon *WWTR1* knockout, but remain largely unchanged when the other genes were knockout (*PIK3CA*, *TP63*, and *SOX2*) (*Figure 6—figure supplement 3A*). We also examined a microarray dataset from *Hiemer et al., 2015* and showed that a significant correlation of *WWTR1* gene expression with *PD-L1* expression was observed, but no correlation in gene expression was seen between *PIK3CA*, *TP63* or *SOX2* and *PD-L1* (*Figure 6—figure supplement 3B*). Together, these results suggested that OSCC resembling the gene signature with WWTR1-dependent cell lines may be associated with better respond to immunotherapy.

## Discussion

New therapeutic targets are urgently needed for the development of OSCC treatment. However, genomics studies have shown that oncogenic mutations in OSCC are largely limited to *PIK3CA* and *HRAS* and even in these, mutations are only found in a small subset of patients. Therefore,

determining the oncogenic pathways and specific therapeutic targets have not been straight forward for this disease. We performed genome-wide CRISPR-Cas9 screens in a unique collection of OSCC cell lines derived from patients with diverse risk habits to identify genetic vulnerabilities that can serve as a basis for further therapeutic development for OSCC. Adding to existing genomics data-sets in OSCC, this approach identifies fitness genes required for the survival of cancer cells where targeting these will result in the killing of these cells. The majority of the cell lines used in this study were from Asian patients where OSCC is more prevalent, and where causative factors include betel quid chewing in addition to smoking that is more often found in Caucasian patients (*Cheong et al., 2017*; *Kumar et al., 2016*). The molecular drivers of these patients are under-characterized as genomics data on Asian patients remain limited (*India Project Team of the International Cancer Genome Consortium, 2013*; *Su et al., 2017*).

Here, we identified 918 fitness genes in OSCC. Pathway enrichment analysis revealed that these genes were highly associated with diverse cancers pathways, confirming the robustness of our screen and pipeline in identifying targetable genetic dependencies. These included known candidate genes that are already being investigated in clinical trials for OSCC or other cancers such as *CDK6*, *PIK3CA*, and *FGFR1*, as well as novel genes that are yet to be explored as therapeutic targets, including those within the oxidative stress pathway (*KEAP1*, *NFE2L2*). Notably, we showed that about 5% (45/918) of these genes are highly tractable with approved drugs, or have drugs that are in late-stage of clinical testing, demonstrating that these screens could help to prioritize drugs that could be repurposed for OSCC treatment. We compared our screens with previous genome-wide RNAi screens and found that candidate genes such as those relating to the cell cycle (*CKAP5, KPNB1, RAN, TPX2*, and *KIF11)* were also identified in our dataset (*Martens-de Kemp et al., 2013*). However, due to stringent filtering of core-fitness genes (*Behan et al., 2019*; *Hart et al., 2014*; *Hart et al., 2017*; *Meyers et al., 2017*), these were no longer within the list of OSCC non-core fitness genes.

Notably, the unique inclusion of several Asian OSCC models known to be associated with betel-quid chewing in our screen enabled us to identify that the NF-kB signaling pathway is among one of the most significantly enriched dependencies among betel-quid associated OSCC compared to cancers not associated with this risk habit. This finding is in line with the past studies reporting direct activation of the NF-kB signaling, upon treatment of OSCC cell lines with the extract of areca nut which is the main component in the betel quid (*Chiang et al., 2008*; *Lin et al., 2005*). By contrast, members of this signaling pathway (*TRIM25, NFKB2, TNFAIP3*) did not seem to be fitness genes in the other cell lines not associated with betel quid chewing, nor those from the Project Score or Dep-Map (*Behan et al., 2019*; *Meyers et al., 2017*). Members of the NF-kB signaling pathways have been proposed as therapeutic targets for inflammatory diseases and cancers, with various types of inhibitors being developed. In the event of the development of NF-kB inhibitosr with promising clinical utililty in the future, it would be of interest to investigate if increase efficacy of this inhibitor would be seen among the betel-quid-associated OSCC. Appreciating this differences in dependencies in Asian OSCC that remains undiscovered in existing large genetic screens could have significant implications, especially when employing precision medicine in the different populations. We acknowledge, however, that further in-depth investigations including a larger sample size and further representative models, are needed in order to confirm our findings and to inform the basis of developing potential targeted therapy against the unique vulnerabilities among the Asian OSCC.

Driver mutations are often expected to be robust biomarkers in precision medicine. However, upon analysing the dependency profiles on cancer genes that are commonly mutated in HNSCC and those with driver mutations among our 21 OSCC cell lines, we show that with the exception of some genes with driver mutations leading to oncogene addiction (*PIK3CA*, *HRAS* and *NFE2L2*), most other driver mutations did not confer preferential gene dependency and their value as a drug target remains unclear.

Given the propensity of copy number alterations in driving OSCC, a 'C' class tumor (*The Cancer Genome Atlas Network, 2015*; *Ciriello et al., 2013*), we investigated the commonly amplified genomic regions to look for functionally important candidate genes. One of the pathways that was significantly enriched was the Hippo pathway. Within this pathway, we focused our analyses on *YAP1* and *WWTR1*, two paralogs that show differential dependency pattern in our 21 OSCC cell lines. Copy number amplifications of 11q22 and 3q25 (where YAP1 and WWTR1 are located, respectively) are common events reported in OSCC, often in mutual exclusive manner (*Campbell et al., 2018*; *Wang et al., 2018*). These two genes are the major effectors negatively regulated by the

Hippo pathway that is increasingly reported to play multiple roles in carcinogenesis, as reviewed comprehensively in recent years (*Dey et al., 2020*; *Santos-de-Frutos et al., 2019*). However, the majority of the studies focused on either one of the paralogs or had assumed similar functions between paralogs (*Santos-de-Frutos et al., 2019*; *Zanconato et al., 2016*). Emerging evidence demonstrate that YAP1 and WWTR1 have distinct roles where they partner with different transcription factors, drive different downstream effectors and also modulate the tumor microenvironment distinctively (*Callus et al., 2019*; *Janse van Rensburg et al., 2018*; *Kaan et al., 2017*; *Plouffe et al., 2018*). Our study revealed the intricate dominance of dependency on either one of the paralogs, despite the other not being deleted or loss. More intriguingly, the subset of lines thought not to be dependent on either YAP1 or WWTR1, were actually lines where the remaining paralog is able to compensate for the lost paralog, therefore, enabling the cells to continue surviving.

In the context of OSCC/HNSCC, studies demonstrating the distinct roles and regulatory mechanism of YAP1 and WWTR1 are emerging. Analysis of genome-wide transcriptional changes upon knockdown of YAP1 or WWTR1 in OSCC showed that YAP1 had a more prominent role in transcriptional regulation (*Hiemer et al., 2015*). Using a tongue orthotopic mouse model with the deletion of MOB1A/B, *Omori et al., 2020* provided strong evidence that YAP1 acted as a strong driver in OSCC tumor initiation and progression, whereby WWTR1 did not seems to play an equivalent role (*Omori et al., 2020*). The differences between YAP1 and WWTR1 can also be further exemplified in terms of their interaction with upstream/downstream pathways, that would involve other frequently co-amplified genes such as *PIK3CA, TP63*, and *SOX2*. The co-occurrence of amplifications in these genes that are part of the extended signaling network of the Hippo pathway underscores the critical role of the Hippo pathway in driving the OSCC tumorigenesis. Overexpression of PIK3CA was shown to be correlated with YAP1 activation and associated with poor clinical outcome (*García-Escudero et al., 2018*). Further, the activation of the PI3K through mitogenic signaling inhibits the Hippo pathway leading to YAP1 activation and cell growth (*Fan et al., 2013*). On the other hand, WWTR1 was shown to act upstream of SOX2, facilitating stemnesses in HNSCC (*Li et al., 2019*). Intriguingly, while WWTR1 knockdown led to reduction of SOX2 mRNA and protein expression, this was not seen when YAP1 was knockdown in the HNSCC cells (Cal27 and Fadu) (*Li et al., 2019*; *Huang et al., 2017*). In support of that, an inverse relationship between the expression levels of YAP1 and ΔNp63 were reported in lung SCC tumor samples (*Huang et al., 2017*). However, this was not seen in HNSCC tumor samples and cell lines (*Ge et al., 2011*), suggesting a cancer/context-specific regulatory mechanism might be in place. Consistent with the oncogenic roles of YAP1 reported in OSCC (*Hiemer et al., 2015*; *Omori et al., 2020*), another study showed that p63, together with the co-expressing chromotin remodeling factor, ACTL6, can drive YAP1 activation, suppressing differentiation and promoting cell proliferation in HNSCC (*Saladi et al., 2017*). Similar observations between WWTR1 and TP63 have not been reported thus far. Hence, understanding the context in which the OSCC lines can be either YAP1-dependent/WWTR1-dependent or having compensable YAP1/WWTR1 is important, as YAP1 and WWTR1-dependency appear to be associated with the enrichment of distinct pathways. Current inhibitors of YAP1 such as verteporfin and CA3 that blocks its interactions with the TEAD transcription factors also targets WWTR1 (*Song et al., 2018*; *Zhang et al., 2015*); therefore, this data underscores the need to develop more specific inhibitors to prevent the targeting of many different downstream pathways. Intriguingly, we also observed enrichment of PIK3CA mutant (p=0.0003) among OSCC lines that are compensable for YAP1 or WWTR1. As recent studies have provided evidence that YAP1 and WWTR1 could mediate mutant PIK3CA-induced tumorigenesis (*Zhao et al., 2018*) andother studies also suggested crosstalk between these Hippo pathway effectors with the PI3K-Akt pathway (*García-Escudero et al., 2018*), confirmatory and mechanistic studies will be needed to delineate why YAP1 and WWTR1 function can be compensated in these *PIK3CA*-mutated cell lines, while distinct dependencies on either paralog are observed in *PIK3CA* wild-type lines. The functional loss of mutated *FAT1* has also been reported to be associated with YAP1 activation in head and neck cancer (*Martin et al., 2018*), however, no enrichment of *FAT1* mutation was seen among the YAP1-dependent nor WWTR1-dependent models in this study.

We also provided tissue-relevant insights of our findings by including analysis of the OSCC tumors from TCGA. Since *YAP1* and *WWTR1* are transcription co-factors that could regulate a plethora of gene transcriptions, we devised an analysis workflow utilizing the DEGs among the three subsets of OSCC models. We have identified 105 OSCC tumors that show highly similar gene

expression signature as the cell lines, which are predicted to share the same dependency pattern. Similarly, comparison of the OSCC tumors based on their gene signatures revealed significant differences in terms of their enriched gene sets and immune signatures. As checkpoint blockade is approved for the treatment of recurrent and metastatic OSCC, understanding how YAP1 and WWTR1 influence the immune microenvironment could provide clues on the combination therapies that could increase the subset of patients responding checkpoint inhibitors. In particular, OSCC with high WWTR1 dependency signature score are significantly associated with various biomarkers that were predicted to show good response toward checkpoint inhibitors. This finding is consistent with the recent discovery that constitutively active WWTR1 induces PD-L1 expression, to a greater extent than YAP1 (*Janse van Rensburg et al., 2018*), and that tumors with YAP1 amplifications have low T-cell infiltration (*Saloura et al., 2019*). These observations have clinical implication as anti-PD1 is an approved therapy for HNSCC and therefore, further investigation and validation will be needed to confirm this observation and its clinical impact. While many companies are developing novel inhibitors targeting YAP1-TEAD transcriptional activity, which should be effective against all other OSCC, combination with checkpoint inhibitor could be considered for those OSCC with WWTR1 dependency signatures. We acknowledge, however, that whilst the dependency observed in cell lines could be recapitulated in OSCC, OSCC tissues would be much more heterogenous and could harbor specific genetic abrogations that could be the dominant driver of tumorigenesis. Therefore, further validation of the association between WWTR1-dependency signature and response to checkpoint inhibitors should be validated particularly in the context of clinical trials involving checkpoint inhibitors.

The roles of the Hippo signaling pathway and its effectors YAP1 and WWTR1 in cancer immunity remains unclear. The inactivation of the Hippo pathway through the loss of LATS1/2 was reported to cause the induction of anti-tumor immune response and inhibition of HNSCC tumor growth, via the hyperactivation of YAP1/WWTR1 (*Barth et al., 2013*; *Moroishi et al., 2016*). This demonstrates that components of the Hippo signaling pathway could also modulate the host tumor microenvironment in addition to what we have demonstrsated in cancer cells. The design and models used in our study have not been set up to examine this where the inherent limitation of using cell lines do not consider the components of the tumor microenvironment in the in vitro screening. Nonetheless, our findings provided a novel insight linking the intricate dependency on YAP1 and WWTR1 with differential state of the immune microenvironment in OSCC which warrants further investigation with the use of immune-competent mouse models, before further clinical evaluation can be made.

In summary, our study is the first large-scale CRISPR-Cas9 screen and focused analysis conducted on large panels of Asian derived OSCC cell lines and provided a cancer-specific overview of the fitness genes landscape, affording opportunities for further therapeutic targets development. The ability to scrutinize the functional genomics of these fitness genes/pathways to a greater detail was also exemplified in this study.

## Materials and methods

### Cell lines

Fourteen OSCC cell lines (referred to as the ORL- series) were derived spontaneously from surgically resected OSCC tissue specimens in Cancer Research Malaysia. Briefly, tissues were collected in α-MEM containing 20% (v/v) FBS, 200 iu/l penicillin, 200 μg/ml streptomycin and 0.1 μg/ml of fungizone. Subsequently, tissues were washed in absolute ethanol for 20–30 s and then washed twice with phosphate-buffered saline (PBS) under sterile conditions. Tissues were minced, washed twice in culture media and re-suspended in α-MEM containing 20% (v/v) FBS, 200 iu/l penicillin, 200 μg/ml streptomycin, 0.4 ng/ml EGF, 2 μg/ml hydrocortisone and 2 mM L-glutamine, and seeded into tissue culture dishes. Cultures were continuously maintained for more than 100 population doublings (*Fadlullah et al., 2016*). HSC-2, HSC-4, and SCC-9 were isolated from squamous cell carcinoma of various oral regions, by which the surgically excised tumors were minced and disaggregated to single cells. Epithelial cells proliferated from the explants were then sub-cultured continuously (*Momose et al., 1989*; *Rheinwald and Beckett, 1981*). No special immortalization methods were detailed for BICR10, PE/CA-PJ15, Ho-1-U-1 and SAS (*Edington et al., 1995*, *Berndt et al., 1997*, *Miyauchi et al., 1985*, *Takahashi et al., 1989*) nor for all other cell lines used in this study. All these

OSCC cells lines were maintained in Dulbecco's Modified Eagle's Medium(DMEM)/Nutrient Mixture F-12 medium (Gibco) supplemented with 10% (v/v) heat inactivated fetal bovine serum (Gibco), and 100 IU Penicillin/Streptomycin (Gibco). All lines were incubated in a humidified atmosphere of 5% CO2 at 37°C. The lines were authenticated by STR profiling using Promega PowerPlex16HS Assay (Promega, Wisconsin, United States), with the data giving more than 80% match to the respective donor or reference as deposited in the databases of cell line resources (such as ATCC, DSMZ, JCRB). Cell lines were routinely tested for the presence of mycoplasma with MycoAlert mycoplasma detection kit (Lonza, Basel, Switzerland). Only mycoplasma-free cell lines were used in all experimentation.

## Plasmid transfection and virus transduction

HEK293 cells were transfected using jetPRIME transfection reagent (Polyplus Transfection) according to the manufacturer's instructions. Briefly, transfection complex consisting of jetPRIME buffer, jet-PRIME reagent, vector of interest, pMD2.G and psPAX2 were prepared and mixed with Opti-MEM (Gibco). Next, the transfection complex medium was added to HEK293 cells with 90% confluency. After overnight incubation at 37°C, 5% $CO_2$, transfection complex medium was replaced with fresh DMEM high glucose (Gibco) complete medium. Medium containing virus was collected at 48- and 72 hr post-transfection and filtered using PVDF 0.45 µm syringe filter.

To perform virus transduction, selected cell lines were transduced with lentivirus containing the vector of interest, in the presence of 8 µg/ml polybrene. After overnight incubation, medium containing lentivirus was replaced with fresh DMEM/F12 complete medium. Cells were incubated for 48 hr and harvested to evaluate the transduction efficiency via flow cytometry analysis with BD LSR Fortessa X-20 cell analyser (BD Biosciences). Gating strategy for flow cytometry analysis of transduced cells carrying fluorescence marker is exemplified in *Supplementary file 8*.

## Generation of Cas9-expressing cell lines

Selected cell lines were transduced with lentivirus containing the pKLV2-EF1α-Cas9Bsd-W (Addgene plasmid # 68343, gift from Kosuke Yusa). Cells stably expressing Cas9 enzyme was established via blasticidin selection 3 days post-transduction. Cas9 enzyme cutting efficiency was routinely checked via lentivirus transduction of the reporter plasmid pKLV2-U6gRNA5(gGFP)-PGKmCherry2AGFP-W (Addgene plasmid # 67982, gift from Kosuke Yusa). The efficiency of Cas9 cutting activity was accessed using a reporter plasmid and analyzed using flow cytometry analysis. The screening will only be conducted on those cell lines with >80% Cas9 cutting activity, as indicated by the efficiency of GFP knockout in cell lines transduced with the reporter plasmid.

## Genome-wide CRISPR-Cas9 knockout screening

The Human Improved Genome-wide Knockout CRISPR Library v1 (Addgene plasmid #67989, gift from Kosuke Yusa) containing 90,709 gRNAs targeting a total number of 18,010 protein-coding genes was used in the genome-wide CRISPR-Cas9 screening (*Tzelepis et al., 2016*). For each of the OSCC cell lines, a total of 60 million Cas9-expressing cells were transduced with the CRISPR library lentivirus at a multiplicity of infection (MOI) of 0.3. Polybrene at 8 µg/ml was added to increase transduction efficiency. All screenings were performed in triplicates. Library representation was evaluated by the percentage of BFP-expressing cells, determined using flow cytometry on day 4 post-transduction. The library representation was minimally kept at 100X coverage of the library, equivalent to 30 million cells total, or 10 million cells expressing BFP before proceeding to puromycin selection (2.0 µg/ml) for 3–4 days, to select for successfully transduced cells. Following complete selection with puromycin, a minimum of 75 million cells were maintained throughout the 18 days screen. BFP expression was monitored to ensure selection was adequate. On day 18 post-transduction, 60 million cells were pelleted down for genomic DNA extraction.

## Genomic DNA extraction of post-CRISPR screened cells

Genomic DNA was extracted from 60 million post-CRISPR screened cells using QIAGEN blood and cell culture DNA Maxi kit (Qiagen), according to manufacturer's instruction. Extracted DNA was quantified using Qubit 2.0 fluorometer (Thermo Fisher Scientific).

## Library preparation of genomic DNA

To prepare and generate Illumina libraries for deep sequencing, amplification of sgRNA was performed using Q5 Hot Start High-Fidelity 2 × Master Mix and forward/reverse primers pair (gLibrary-HiSeq_50bp-SE-U1 F and R) in 50 μl reaction, as previously described[4]. For the CRISPR library v1 plasmid, 10 independent PCR reactions were set up using 2.0 μg of the plasmid. While for the CRISPR-screen cell lines, 2.0 μg of genomic DNA harvested from day-18 post-transduction was used in each of the 36 independent PCR reactions. The PCR conditions were as follows: 98°C for 30 s, 26–28 cycles of 98°C for 10 s, 61°C for 15 s and 72°C for 20 s, and the final extension, 72°C for 2 min. The PCR products were analyzed on 2% agarose gel and additional PCR cycles were added if necessary.

About 5 μl of PCR products were pooled from all 36 reactions and QIAquick PCR purification kit (QIAGEN) was used to purify the amplified gRNA. Concentration of purified PCR products were quantified with Qubit dsDNA broad-range (BR) assay kit (Thermo Fisher Scientific), using the Qubit 2.0 fluorometer (Thermo Fisher Scientific). PCR enrichment was then carried out using 200 pg of purified PCR products with 2x KAPA HotStart ReadyMix. 1 μl of forward P5 fusion primer (PE 1.0 p5 Top_PE_C) and 1 μl of different reverse primers (indexed iPCRTags) were used. The PCR conditions were as follows: 98°C for 30 s, 10–12 cycles of 98°C for 10 s, 66°C for 15 s and 72°C for 20 s, and the final extension, 72°C for 5 min. Finally, SPRISelect beads (Beckman Coulter) were used to purify the PCR products at a PCR-product-to-bead ratio of 1:0.8. Purified libraries were dissolved in 30 μl nuclease-free water and quantified using Agilent High Sensitivity DNA kit (Agilent Technologies) on Agilent 2100 Bioanalyzer (Agilent Technologies).

## Deep sequencing of post-CRISPR screen libraries

Purified libraries of the triplicate screens tagged with different iPCR tags were pooled and sequenced at about 300x coverage on Illumina HiSeq 2500 with 19 bp single-end (SE) deep sequencing at the Wellcome Sanger Institute (WSI). About 30–40 million reads were obtained for each of the three replicates. Sequences of the Read one sequencing primer (U6-Illumina-seq2) and index sequencing primer can be found in *Supplementary file 9*.

## CRISPR data processing

sgRNA raw counts in each triplicate of the CRISPR screen were generated using the in-house script developed at WSI, as previously described (*Behan et al., 2019*; *Iorio et al., 2018*). The CRISP-RcleanR tool was downloaded as an R package (https://github.com/francescojm/CRISPRcleanR) and used for pre-processing of the sgRNA raw counts (*Iorio et al., 2018*). This tool allows for an unsupervised correction for copy number amplification bias and other gene-independent responses when subjected to CRISPR-Cas9 targeting, hence reducing false-positive call for essential genes (*Iorio et al., 2018*). Briefly, the *ccr.NormfoldChanges* function was used to compute the median-ratio normalization of raw counts and log2 fold-changes for all sgRNAs, averaging from triplicates. The built-in KY_library_v1.0 was used for library annotation and sgRNAs with read counts less than 30 in the plasmid were excluded. Then, the *ccr.logFCs2chromPos* and *ccr.GWclean* function of the CRISPRcleanR were used to perform genome mapping and sorting of the sgRNAs, followed by the correction of the gene-independent responses to compute corrected log-fold changes. As part of the quality assessment, Pearson correlation test was used to compare sgRNA raw counts between replicates of the same cell line, while precision-recall was assessed as previously described, using sets of known essential and non-essential genes (*Behan et al., 2019*; *Supplementary file 10*).

The corrected gene-level log-fold changes were quantile normalized and corrected for batch effect using ComBat (*Leek et al., 2012*), we refer this CRISPRcleanR corrected, quantile normalized and ComBat corrected log-fold changes as the 'CRISPR score', with a negative value indicating the extent of depletion of sgRNA counts targeting the gene when compared with the initial plasmid library.

The function *ccr.correctCounts* used inverse-transformation method to generate CRISPRcleanR-corrected counts, which is used as input files for MAGeCK analysis (*Iorio et al., 2018*; *Li et al., 2014*). For each of the 21 OSCC cell lines, MAGeCK analysis was performed using default parameters, except that normalization is set to 'none', as the input corrected counts had already been

normalized using CRISPRcleanR. A false discovery rate cut-off of 5% (FDR $\leq$ 0.05) was applied to identify the significantly depleted genes in each cell line, defined here as MAGeCK hits.

To remove potential false-positive hits, RNA-seq expression data of each of the 21 cell lines were utilized to filter out MAGeCK hits with negligibly low or no reported reads (Fragments Per Kilobase of transcript per Million mapped reads (FPKM) <0.5).

Next, in order to identify and prioritize genes that can be safely targeted for the treatment of OSCC, we curated a list of core fitness essential genes from four different sources and used it to further filter the list of MAGeCK hits from each cell line. The first two sources were the core essential genes (CEG) list, published by *Hart et al., 2014* and the subsequent updated CEG2 list, published in 2017 (*Hart et al., 2017*). The third source is the 'common-essential genes', downloaded from Broad's Institute Cancer Dependency Map database (18Q3 release) (*Meyers et al., 2017*). We also utilized the list of pan-cancer core-fitness genes compiled by Project Score of Cancer Dependency Map at WSI (*Behan et al., 2019*). The full list of all genes from these four sources was tabulated in *Supplementary file 2*.

## WES and identification of driver mutation

All 21 OSCC cell lines were subjected to WES at the WSI using HiSeq2500. WES data were processed using an established pipeline as previously described to identify driver mutations (*Iorio et al., 2016*). A total of 43 genes with driver mutation in at least one cell line were identified (*Supplementary file 5*).

## Pathway enrichment analysis

KEGG pathway enrichment analysis was performed using the over-representation analysis function at the ConsensusPathDB (http://ConsensusPathDB.org) (*Kamburov et al., 2013*). A threshold of minimum two genes overlapping with the gene set of a given pathway and p-value cut-off of 0.05 were applied. Enriched pathways were ranked by q-value.

## Differential fitness genes analysis for betel-quid-associated OSCCs

OSCC models with known betel quid chewing as the only risk habit were included in this analysis – (n = 7; ORL-115, ORL-136, ORL-174, ORL-195, ORL-204, ORL-207 and ORL-214), to compare with the other OSCCs not associated with betel quid chewing (n = 14; ORL-48, ORL-150, ORL-153, ORL-156, ORL-166, ORL-188, ORL-215, BICR10, Ho-1-u-1, HSC-2, HSC-4, SAS, SCC-9, PE/CA-PJ15). Venn diagram was used to depict the number of unique and overlapping fitness genes that are found between those OSCCs with or without association with betel quid chewing.

## Functional classification and tractability assessment

To assess the tractability of the 918 non-core fitness genes, we utilized the genome-wide target tractability assessment pipeline as previously described (*Behan et al., 2019*; *Brown et al., 2018*). Based on assessment for small molecules tractability, essential genes were assigned into tractability bucket 1 to 10, with decreasing tractability. Next, essential genes in each tractability group were further classified into protein classes using the PANTHER database online tool (http://www.pantherdb.org/) (*Mi et al., 2013*).

## Validation sgRNA design and cloning

To validate the results obtained from the screen, individual targeted genetic knockouts were generated using CRISPR/Cas9 and plasmid expressing sgRNAs targeting the gene of interest. Two sgRNA sequences were used for each target gene, one was selected from the CRISPR library v1 while another sequence was designed using the Genetic Perturbation Platform (GPP) web portal (https://portals.broadinstitute.org/gpp/public/analysis-tools/sgrna-design). List of sgRNA used and their sequences can be found in *Supplementary file 11*. pKLV2-U6gRNA5(BbsI)-PGKpuro2ABFP-W and pKLV2-U6gRNA5(BbsI) PGKpuro2AmCherry-W were linearized using *BbsI* enzyme (NEB R0539S) and the concentration was adjusted to 20 ng/μL. Target oligos were phosphorylated and annealed using T4 PNK (NEB M0201). The thermocycler condition used are as follows: 37°C for 30 min, 95°C for 5 min, followed by a ramp down to 25°C at 0.1°C /s. The annealed oligos were next diluted twice for prior to ligation: 1st dilution = 139 μL EB buffer + 2 μL of 10 μM double-stranded oligos; 2nd

dilution = 57 µL EB buffer + 3 µL of 1st dilution. Following that, overnight ligation at 4°C was carried out using T4 ligase (NEB M0202S) and 10X ligase buffer (NEB M0202S). The ligation products were then transformed into DH5α chemically- generated competent cells and plated onto Luria Broth (LB) agar plates containing 100 µg/ml Ampicillin. The plasmids were then extracted using QIAprep Spin Miniprep Kit (QIAGEN) and the sgRNA sequences were verified by Sanger sequencing prior to use.

## Co-competition assay

The relative growth rate of sgRNA-transduced and non-transduced cells was compared using co-competiton assay, as described previously (*Behan et al., 2019*; *Tzelepis et al., 2016*). Briefly, in order to achieve single gene-specific knockout, the Cas9-expressing cell lines were transduced at 30–70% transduction efficiency, with lentivirus carrying gene-specific sgRNA in pKLV2-U6gRNA5 (BbsI)-PGKpuro2ABFP-W. Using flow cytometry, the percentage of BFP-positive sgRNA-transduced cells was measured between day 4 and day 18 post-transduction. The results obtained from days 8, 11, 15 to 18 were normalized to the percentage of BFP-positive transduced cells on day 4 or 6 to investigate the relative growth changes of the transduced population following gene depletion. For each target gene (*YAP1* and *WWTR1*), two different sgRNA were used, one from the Kosuke Yusa's CRISPR Library v1 ('Y1K' – sgRNA targeting YAP1; 'W1K – sgRNA targeting WWTR1') and another independently designed sgRNA using Broad's sgRNA-designer tool ('Y2B' – sgRNA targeting YAP1; 'W2B' – sgRNA targeting WWTR1). sgRNA targeting a core fitness gene, Polo-like kinase 1 (*PLK1*) was included as a positive control, choline acetyltransferase (*CHAT*)-targeting sgRNA was used as a non-fitness gene negative control and a non-targeting (NT) sgRNA w also included.

To achieve double gene knockout, sgRNAs targeting two different genes were cloned into either one of the plasmids tagged with BFP or mCherry (pKLV2-U6gRNA5(BbsI)-PGKpuro2ABFP-W or pKLV2-U6gRNA5(BbsI)-PGKpuro2AmCherry-W). Thereafter, the changes in the BFP- and mCherry double-positive cell population was measured as mentioned above.

## Lysates preparation and western blotting

To determine the baseline protein expression level, OSCC parental cell lines were seeded in 100 mm$^3$ dish and cultured until they reached 70–80% confluency. To assess the differential expression of the protein of interest after gene knockdown, Cas9-expressing cells were transduced with the target sgRNA at above 90% transduction efficiency. Next day, transduced cells were selected using 2 µg/ml of Puromycin. Day four post-transduction, percentage of BFP-expressing cells were determined using flow cytometry and total cell lysates (TCL) were extracted with RIPA buffer (50 mM Tris pH8, 1% (v/v) NP-40, 0.5% (w/v) sodium deoxycholate, 0.1% (w/v) SDS, 150 mM NaCl) supplemented with Halt Protease and Phosphatase Inhibitor (PI) Cocktail (Pierce Biotechnology) on ice. TCL was collected by centrifugation and quantified using the BCA method (Thermo Fisher Scientific). About 20 µg of the TCL was resolved on SDS-PAGE gel and proteins were transferred onto PVDF membranes (Millipore). Membranes were blocked with 5% (w/v) milk in TBST (0.1% (v/v) Tween 20) and probed with primary antibodies (1:1000 dilution in 1% (w/v) BSA) overnight at 4°C. Horseradish peroxidase (HRP)-conjugated secondary antibodies (1:10,000 dilution in 5% milk) were probed for one hour at room temperature. For signal development, WesternBright Quantum HRP substrate (Advansta Inc) was used and visualized using the FluorChem HD2 imaging systems (Alpha Innotech). To normalize for loading, the blots were re-probed with an anti-tubulin monoclonal antibody (1:1000 dilution in 1% BSA) and processed as described above. List of primary and secondary antibodies used is found in *Supplementary file 12*. Uncropped western blot images can be found in *Supplementary file 13*.

## Total RNA extraction and qPCR

Cas9-expressing cells were transduced with selected sgRNA to achieve single gene or double gene knockout, as described above. On day 4 post-transduction, total RNA was extracted using TRIzol Reagent (Thermo Fisher Scientific). Total RNA (1 µg) was used for reverse transcription to complementary DNA (cDNA) using high-capacity cDNA reverse transcription kit (Applied Biosystems). Real-time quantitative PCR was performed using 1 µl of 5x diluted cDNA with PowerUp SYBR Green Master Mix and corresponding primers in 7500 Real-Time PCR System (Applied Biosystem). All reactions were performed in technical triplicates and repeated twice. Cycling conditions used are as follows:

50°C for 2 min, 95°C for 2 min, 40 cycles of 95°C for 15 s, and 60°C for 1 min. A default melt curve stage was included to allow inspection of primer specificity. Ribosomal protein L13 (RPL13) was used as an endogenous reference control for normalization. Sequences of all primers used can be found in *Supplementary file 9*.

## Clonogenic assay

On 4- or 6 days post-transduction of sgRNA-containing lentiviruses, 2000 cells were seeded into six-well plate. After a week, cells were fixed using ice-cold methanol followed by staining with crystal violet solution.

## Cell viability assay

The 3-(4,5-dimethylthiazol-2-yl)−2,5-diphenyltetrazolium bromide (MTT) assay was used to access the effect of target gene knockdown/knockout on cell viability. Briefly, 4- or 6 days post-transduction with sgRNA, 2000 cells were seeded in triplicates in 96-well plate. 72 hr later, 50 µl MTT was added to each well and incubated for 4 hr at 37°C. After removing the media, 150 µl dimethyl sulfoxide was added to dissolve the formazan crystal and optical density was measured using Synergy H1M micro-plate reader (BioTek Instruments, USA) at 570 nm.

## Apoptosis assay

On 4- or 6 days post-transduction of sgRNA-containing lentiviruses, 30,000 cells were seeded into 24-well plate and harvested after 72 hr. Cells were pelleted down and washed with PBS. For detection of apoptotic cells, the cell pellet was resuspended in 1x Annexin V buffer containing 2.5 µl of Annexin V solution (BD Biosciences) and 2.5 µl of propidium iodide and incubated for 15 min in the dark. The proportion of apoptotic cells were analyzed using the LSR Fortessa X-20 cell analyser (BD Biosciences) and FlowJo (version 10.5.3, BD Biosciences), considering all single- and double-stained cells as apoptotic cells. Gating strategy for detection of apoptotic cells using flow cytometry can be found in *Supplementary file 8*.

## DEGs signature based on dependency on YAP1 or WWTR1

Three representative cell lines with validated dependency on YAP1 or WWTR1 were used to derive gene expression signatures. 'YAP1-dependent' – ORL-48, ORL-204, SAS; 'WWTR1-dependent' – ORL-214, PE/CA-PJ15, ORL-174; 'Compensable' – BICR10, HSC-2 and HSC-4. DEGs for each group were computed using the limma package (Bioconductor) on the iRAP-processed, ComBat corrected FPKM matrix for these nine cell lines. Non-overlapping DEGs with significant p-value threshold <0.01 and log fold change >2 were retained. The final list of DEGs is found in *Supplementary file 6*. Gene expression data of the HNSCC cohort in TCGA in the form of RSEM was downloaded from cbioportal (https://www.cbioportal.org/) (*Gao et al., 2013*). Gene expressions of all DEGs were then extracted for the 315 OSCC samples.

For all DEGs, Z-score was computed and a 'dependency signature score'/ 'compensable signature score' was generated for each cell line/tumor sample, taking the difference between the average of all Z-score of upregulated DEGs and that of downregulated DEGs. For example, 'YAP1-dependency signature score' = (average of all Z-score of upregulated DEGs among YAP1-dependent cell lines) - (average of all Z-score of downregulated DEGs among YAP1-dependent cell lines).

Subsequently, cell lines and OSCC tumors were analyzed using hierarchical clustering and visualize with heatmap (generated using Morpheus, Broad Institute: https://software.broadinstitute.org/morpheus) using the 'YAP1 dependency signature score', 'WWTR1 dependency signature score' and 'Compensable signature score'.

## Correlation analysis and gene set enrichment analysis

Core OSCC samples with >0.5 dependency signature score in one of the three groups were identified. There was a total of 43 OSCC samples with high YAP1 dependency signature score; 31 with high WWTR1 dependency signature score and 30 with high Compensable signature score. Clinical and genomic data of these core OSCC samples were accessed from cBioPortal (*Gao et al., 2013*). Gene set enrichment analyses (GSEA) for the three groups of core OSCC samples were performed using the Broad Institute's Molecular Signatures Database (MSigDB) hallmark gene sets as reference

database (*Liberzon et al., 2015*). Signatures reflective of the immune landscape of these core OSCC samples were extracted from the supplementary table 1 of *Thorsson et al., 2018*. Single-sample GSEA (ssGSEA) was performed using the GenePatterns web-tool (https://www.genepattern.org/) (*Reich et al., 2006*) with the 18-genes T-cell inflamed Gene Expression Profile (GEP) gene set, which was found to be predictive biomarkers for response to pembrolizumab in HNSCC clinical trial (*Ayers et al., 2017*; *Cristescu et al., 2018*).

## Statistical analysis

All statistical significance analyses were performed using unpaired parametric two-tailed t-test in GraphPad Prism (version 8, GraphPad Software Inc) unless otherwise stated. Unpaired t-test with Welch's correction (Welch's t-test) was used for all analyses in *Figure 6* and *Figure 6—figure supplement 2* due to unequal sample size. For estimation of the Pearson correlation, the *cor.test* function in Rstudio (version 1.2.1335, Rstudio Inc) was used.

## Code availability

No unreported or custom code was used in this study. Open source softwares were used for data analysis and codes are available upon request.

## Acknowledgements

We thank Dr. Kosuke Yusa for providing the CRISPR Library v1 and Dr. Iuan Sheau Chin for technical assistance in propagating the CRISPR Library v1. This work was supported by the Newton-Ungku Omar Fund and the Medical Research Council, United Kingdom (MR/P013457/1) and other sponsors of Cancer Research Malaysia. Work in MJG lab is supported by Wellcome (206194). Cancer Research Malaysia is a non-profit research organization, committed to an understanding of cancer prevention, diagnosis and treatment through a fundamental research program.

## Additional information

### Competing interests

Ultan McDermott: is affiliated with AstraZeneca. The other authors declare that no competing interests exist.

### Funding

| Funder | Grant reference number | Author |
| --- | --- | --- |
| Newton-Ungku Omar Fund and Medical Research Council, United Kingdom | MR/P013457/1 | Ultan McDermott<br>Mathew J Garnett<br>Sok Ching Cheong |
| Wellcome Trust | 206194 | Stacey Price<br>Emanuel Gonçalves<br>Fiona M Behan<br>Jessica Bateson<br>James Gilbert<br>Mathew J Garnett |
| Cancer Research Malaysia | | Annie Wai Yeeng Chai<br>Pei San Yee<br>Shi Mun Yee<br>Hui Mei Lee<br>Vivian KH Tiong<br>Sok Ching Cheong |

This work was directly supported by the Newton-Ungku Omar Fund and the Medical Research Council, United Kingdom (MR/P013457/1). Authors in SCC's group were supported by sponsors of Cancer Research Malaysia; Authors in MJG's group were supported by Wellcome. Other funders had no role in study design, data collection and interpretation, or the decision to submit the work for publication.

## Author contributions
Annie Wai Yeeng Chai, Conceptualization, Data curation, Formal analysis, Validation, Investigation, Visualization, Methodology, Writing - original draft, Project administration, Writing - review and editing; Pei San Yee, Shi Mun Yee, Data curation, Formal analysis, Validation, Investigation, Visualization, Methodology, Writing - original draft, Writing - review and editing; Stacey Price, Data curation, Formal analysis, Validation, Investigation, Methodology, Project administration, Writing - review and editing; Hui Mei Lee, Vivian KH Tiong, Data curation, Formal analysis, Investigation, Methodology, Writing - review and editing; Emanuel Gonçalves, Data curation, Software, Formal analysis, Investigation, Visualization, Methodology, Writing - review and editing; Fiona M Behan, Formal analysis, Investigation, Methodology, Writing - review and editing; Jessica Bateson, Data curation, Investigation, Methodology; James Gilbert, Data curation, Software, Formal analysis, Methodology; Aik Choon Tan, Conceptualization, Visualization, Methodology, Writing - review and editing; Ultan McDermott, Mathew J Garnett, Conceptualization, Resources, Supervision, Funding acquisition, Project administration, Writing - review and editing; Sok Ching Cheong, Conceptualization, Resources, Supervision, Funding acquisition, Writing - original draft, Project administration, Writing - review and editing

## Author ORCIDs
Annie Wai Yeeng Chai 🆔 https://orcid.org/0000-0002-8015-6050
Emanuel Gonçalves 🆔 http://orcid.org/0000-0002-9967-5205
Sok Ching Cheong 🆔 https://orcid.org/0000-0002-6765-9542

## Decision letter and Author response
Decision letter https://doi.org/10.7554/eLife.57761.sa1
Author response https://doi.org/10.7554/eLife.57761.sa2

# Additional files
## Supplementary files
• Supplementary file 1. Number of MAGeCK defined fitness genes and percentage overlap with core fitness gene.

• Supplementary file 2. List of core fitness genes from four different sources.

• Supplementary file 3. KEGG pathway analysis results for fitness genes. (A) KEGG pathway analysis results for all 2539 essential genes identified from CRISPR screen of 21 OSCC cell lines. (B) KEGG pathway analysis results for 918 essential genes after filtering out core fitness genes.

• Supplementary file 4. Details of driver mutations in 43 genes identified from whole exome sequencing on 21 OSCC cell lines.

• Supplementary file 5. Classification of the 918 non-core essential genes based on target tractability.

• Supplementary file 6. Differentially expressed genes (DEGs) used to derived Z-score for computing of dependency scores.

• Supplementary file 7. GSEA enrichment analysis on cancer hallmarks for cell lines and OSCC tumors.

• Supplementary file 8. Representative figures exemplifying gating strategies in flow cytometry analysis.

• Supplementary file 9. List of primers.

• Supplementary file 10. Quality assessment of the genome-wide CRISPR-Cas9 screen.

• Supplementary file 11. List of sgRNA and their sequences.

• Supplementary file 12. List of antibodies.

• Supplementary file 13. All uncropped western blot images.

• Transparent reporting form

## Data availability

All main data generated or analysed during this study are included in the manuscript and supplementary files. Source data files for each figures and supplements have also been provided. The larger datasets of CRISPR screens, WES and RNA-sequencing output are available from Figshare (https://doi.org/10.6084/m9.figshare.11919753).

The following dataset was generated:

| Author(s) | Year | Dataset title | Dataset URL | Database and Identifier |
|---|---|---|---|---|
| Annie WYC, PSY, SP, SMY, HML, VKHT, EG, FB, JB, JG, ACT, UMD, MJG, SCC | 2020 | Genome-wide CRISPR screens reveal fitness genes in the Hippo pathway for oral squamous cell carcinoma | https://doi.org/10.6084/m9.figshare.11919753 | figshare, 10.6084/m9.figshare.11919753 |

The following previously published dataset was used:

| Author(s) | Year | Dataset title | Dataset URL | Database and Identifier |
|---|---|---|---|---|
| Cance Genome Atlas Network | 2015 | Head and Neck Squamous Cell Carcinoma (TCGA, Nature 2015) | http://www.cbioportal.org/study/summary?id=hnsc_tcga_pub | cbioportal, hnsc_tcga_pub |

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

# Appendix 1

## Key Resources Table

**Appendix 1—key resources table**

| Reagent type (species) or resource | Designation | Source or reference | Identifiers | Additional information |
|---|---|---|---|---|
| Recombinant DNA reagent | Human Improved Genome-wide Knockout CRISPR Library v1 (pooled library) | Addgene | #67989 | Containing 90,709 gRNAs, selectable with puromycin |
| Recombinant DNA reagent | pKLV2-EF1α-Cas9Bsd-W (plasmid) | Addgene | #68343 | To express Cas9 constitutively, selectable with blasticidin |
| Recombinant DNA reagent | pKLV2-U6gRNA5 (gGFP)-PGKmCherry2AGFP-W (plasmid) | Addgene | #67982 | Reporter plasmid |
| Recombinant DNA reagent | pKLV2-U6gRNA5(BbsI)-PGKpuro2ABFP-W (plasmid) | Addgene | #67974 | For cloning of single sgRNAs, tag with blue fluorescence protein (BFP) |
| Recombinant DNA reagent | pKLV2-U6gRNA5(BbsI)-PGKpuro2AmCherry-W (plasmid) | Addgene | #67977 | For cloning of single sgRNAs, tag with mcherry |
| Recombinant DNA reagent | pMD2.G (plasmid) | Addgene | #12259 | For lentivirus generation |
| Recombinant DNA reagent | psPAX2 (plasmid) | Addgene | #12260 | For lentivirus generation |
| Chemical compound, drug | jetPRIME reagent and buffer | Polyplus transfection | 114–75 | For transfection |
| Cell line (*Homo sapiens*) | Cells derived from oral squamous cell carcinoma | Cancer Research Malaysia *Fadlullah et al., 2016*, PMID:27050151 | ORL-48 | Maintained in DMEM/F12 supplemented with 10% FBS, 1% penicillin/streptomycin |
| Cell line (*Homo sapiens*) | Cells derived from oral squamous cell carcinoma | Cancer Research Malaysia *Fadlullah et al., 2016*, PMID:27050151 | ORL-115 | Maintained in DMEM/F12 supplemented with 10% FBS, 1% penicillin/streptomycin |
| Cell line (*Homo sapiens*) | Cells derived from oral squamous cell carcinoma | Cancer Research Malaysia *Fadlullah et al., 2016*, PMID:27050151 | ORL-136 | Maintained in DMEM/F12 supplemented with 10% FBS, 1% penicillin/streptomycin |
| Cell line (*Homo sapiens*) | Cells derived from oral squamous cell carcinoma | Cancer Research Malaysia *Fadlullah et al., 2016*, PMID:27050151 | ORL-150 | Maintained in DMEM/F12 supplemented with 10% FBS, 1% penicillin/streptomycin |
| Cell line (*Homo sapiens*) | Cells derived from oral squamous cell carcinoma | Cancer Research Malaysia *Fadlullah et al., 2016*, PMID:27050151 | ORL-153 | Maintained in DMEM/F12 supplemented with 10% FBS, 1% penicillin/streptomycin |
| Cell line (*Homo sapiens*) | Cells derived from oral squamous cell carcinoma | Cancer Research Malaysia *Fadlullah et al., 2016*, PMID:27050151 | ORL-156 | Maintained in DMEM/F12 supplemented with 10% FBS, 1% penicillin/streptomycin |
| Cell line (*Homo sapiens*) | Cells derived from oral squamous cell carcinoma | Cancer Research Malaysia *Fadlullah et al., 2016*, PMID:27050151 | ORL-166 | Maintained in DMEM/F12 supplemented with 10% FBS, 1% penicillin/streptomycin |
| Cell line (*Homo sapiens*) | Cells derived from oral squamous cell carcinoma | Cancer Research Malaysia *Fadlullah et al., 2016*, PMID:27050151 | ORL-174 | Maintained in DMEM/F12 supplemented with 10% FBS, 1% penicillin/streptomycin |

*Continued on next page*

*Appendix 1—key resources table continued*

| Reagent type (species) or resource | Designation | Source or reference | Identifiers | Additional information |
|---|---|---|---|---|
| Cell line (*Homo sapiens*) | Cells derived from oral squamous cell carcinoma | Cancer Research Malaysia *Fadlullah et al., 2016*, PMID:27050151 | ORL-188 | Maintained in DMEM/F12 supplemented with 10% FBS, 1% penicillin/streptomycin |
| Cell line (*Homo sapiens*) | Cells derived from oral squamous cell carcinoma | Cancer Research Malaysia *Fadlullah et al., 2016*, PMID:27050151 | ORL-195 | Maintained in DMEM/F12 supplemented with 10% FBS, 1% penicillin/streptomycin |
| Cell line (*Homo sapiens*) | Cells derived from oral squamous cell carcinoma | Cancer Research Malaysia *Fadlullah et al., 2016*, PMID:27050151 | ORL-204 | Maintained in DMEM/F12 supplemented with 10% FBS, 1% penicillin/streptomycin |
| Cell line (*Homo sapiens*) | Cells derived from oral squamous cell carcinoma | Cancer Research Malaysia *Fadlullah et al., 2016*, PMID:27050151 | ORL-207 | Maintained in DMEM/F12 supplemented with 10% FBS, 1% penicillin/streptomycin |
| Cell line (*Homo sapiens*) | Cells derived from oral squamous cell carcinoma | Cancer Research Malaysia *Fadlullah et al., 2016*, PMID:27050151 | ORL-214 | Maintained in DMEM/F12 supplemented with 10% FBS, 1% penicillin/streptomycin |
| Cell line (*Homo sapiens*) | Cells derived from oral squamous cell carcinoma | Cancer Research Malaysia *Fadlullah et al., 2016*, PMID:27050151 | ORL-215 | Maintained in DMEM/F12 supplemented with 10% FBS, 1% penicillin/streptomycin |
| Cell line (*Homo sapiens*) | Cells derived from oral squamous cell carcinoma | Wellcome Sanger Institute | BICR10 | Maintained in DMEM/F12 supplemented with 10% FBS, 1% penicillin/streptomycin |
| Cell line (*Homo sapiens*) | Cells derived from oral squamous cell carcinoma | Wellcome Sanger Institute | HO-1-u-1 | Maintained in DMEM/F12 supplemented with 10% FBS, 1% penicillin/streptomycin |
| Cell line (*Homo sapiens*) | Cells derived from oral squamous cell carcinoma | Wellcome Sanger Institute | HSC-2 | Maintained in DMEM/F12 supplemented with 10% FBS, 1% penicillin/streptomycin |
| Cell line (*Homo sapiens*) | Cells derived from oral squamous cell carcinoma | Wellcome Sanger Institute | HSC-4 | Maintained in DMEM/F12 supplemented with 10% FBS, 1% penicillin/streptomycin |
| Cell line (*Homo sapiens*) | Cells derived from oral squamous cell carcinoma | Wellcome Sanger Institute | PE/CA-PJ15 | Maintained in DMEM/F12 supplemented with 10% FBS, 1% penicillin/streptomycin |
| Cell line (*Homo sapiens*) | Cells derived from oral squamous cell carcinoma | Wellcome Sanger Institute | SAS | Maintained in DMEM/F12 supplemented with 10% FBS, 1% penicillin/streptomycin |
| Cell line (*Homo sapiens*) | Cells derived from oral squamous cell carcinoma | Wellcome Sanger Institute | SCC-9 | Maintained in DMEM/F12 supplemented with 10% FBS, 1% penicillin/streptomycin |
| Cell line (*Homo sapiens*) | Human embryonic kidney cells | Wellcome Sanger Institute | HEK293 | Maintained in DMEM (high glucose) supplemented with 10% FBS, 1% penicillin/streptomycin |
| Antibody | Anti-human YAP (D8H1X), Rabbit monoclonal | Cell Signaling Technology | #14074 | WB - (1:1000) |
| Antibody | Anti-human TAZ (D361D), Rabbit monoclonal | Cell Signaling Technology | #70148 | WB - (1:1000) |
| Antibody | Anti-human actin (clone C4), Mouse monoclonal | Merck Milipore | MAB1501 | WB – (1:5000) |
| Antibody | Goat anti-rabbit IgG-HRP | Southern Biotec | SB4010-05 | Secondary antibody; WB – (1:10000) |
| Antibody | Goat anti-mouse IgG-HRP | Southern Biotec | SB1010-05 | Secondary antibody; WB – (1:10000) |

*Continued on next page*

*Appendix 1—key resources table continued*

| Reagent type (species) or resource | Designation | Source or reference | Identifiers | Additional information |
|---|---|---|---|---|
| Commercial assay or kit | Q5 Hot Start High-Fidelity 2X Master Mix | New England Biolabs | M0494S | |
| Commercial assay or kit | QIAquick PCR purification kit | Qiagen | 28104 | |
| Commercial assay or kit | Qubit dsDNA broad-range (BR) assay kit | Thermo Fisher Scientific | Q32850 | |
| Commercial assay or kit | KAPA HotStart ReadyMix | Roche | KK2601 | |
| Commercial assay or kit | Agilent High Sensitivity DNA kit | Agilent technologies | 5067–4626 | |
| Commercial assay or kit | QIAprep Spin Miniprep Kit | Qiagen | 27104 | |
| Commercial assay or kit | cDNA reverse transcription kit | Applied Biosystems, Thermo Fisher Scientific | 4368813 | |
| Commercial assay or kit | PowerUp SYBR Green Master Mix | Applied Biosystems, Thermo Fisher Scientific | A25742 | |
| Commercial assay or kit | FITC Annexin V Apoptosis Detection Kit I | BD Bioscience | 556547 | |
| Chemical compound, drug | BbsI enzyme | New England Biolabs | R0539S | |
| Chemical compound, drug | T4 Polynucleotide kinase enzyme | New England Biolabs | M0201 | |
| Chemical compound, drug | T4 ligase and 10x ligase buffer | New England Biolabs | M0202S | |
| Chemical compound, drug | WesternBright Quantum HRP substrate | Advansta | K-12042-D20 | Substrate for Western Blot |
| Chemical compound, drug | TRIzol Reagent | Thermo Fisher Scientific | 15596026 | |
| Software, algorithm | CRISPRcleanR (version 0.5) | *Iorio et al., 2018* PMID:30103702 | | |
| Software, algorithm | MAGeCK (version 0.5.7) | *Li et al., 2014* PMID:25476604 | | |
| Software, algorithm | Rstudio (version 1.2.1335) | Rstudio Inc | | |
| Software, algorithm | GraphPad Prism (version 8) | GraphPad Software Inc | | |

