## [Decision Letter]

**Acceptance summary:**

This work defines genes required for survival and robust growth by oral squamous cell carcinoma (OSCC) cells in Western and Asian populations. Following a genome-wide CRISPR screen, the authors identified components of the Hippo/Yap pathway as important for growth of multiple OSCC cell lines. In dissecting the pathway, the authors determined that some OSCC cell lines depend more on YAP1, some more on WWTR1/TAZ, while some depend on both, suggesting more heterogeneity in response than was previously appreciated. Importantly, the work also suggests some specific features of transcriptional pathways, including activation of the NF-κB pathway, in OSCC derived from Asians with a habit of betel quid chewing. These different dependencies may have clinical consequences, as the data suggests that OSCC cells specifically dependent on WWTR1 are associated with a distinct transcriptome and immune signature associated with response to immunotherapy.

**Decision letter after peer review:**

Thank you for submitting your article "Genome-wide CRISPR screens of oral squamous cell carcinoma reveal fitness genes in the Hippo pathway" for consideration by *eLife*. Your article has been reviewed by two peer reviewers, and the evaluation has been overseen by a Reviewing Editor and Päivi Ojala as the Senior Editor. The following individual involved in review of your submission has agreed to reveal their identity: Marius Sudol (Reviewer #2).

The reviewers have discussed the reviews with one another and the Reviewing Editor has drafted this decision to help you prepare a revised submission.

Summary:

This manuscript by Annie Chai and colleagues from the laboratory of Sok Ching Cheong reports in the CRISPR/Cas9-based screen of cells derived from oral squamous cell carcinoma (OSCC), which were predominantly of Asian, Malaysian origin. The screen was aimed at the identification of "fitness genes". Several of these genes, some known and some new, were given non-essential status and a new set of essential genes was identified as worthwhile targets of a potential therapy. The authors identified components of the Hippo/Yap pathway as important for growth of multiple OSCC cell lines. For more than a decade, the two main pro-oncogene effectors of the Hippo pathway, namely YAP and TAZ, have been considered acting in unison in cancers. The current report indicates that YAP and TAZ may play specific roles in cancer. In dissecting the pathway, the authors determined that some OSCC cell lines depend more on YAP1, some more on WWTR1/TAZ, and in some either protein can complement the function of the other, suggesting more heterogeneity in response than was previously appreciated. The copy number amplification of WWTR1/TAZ was shown as likely to constitute a functional oncogenic role of this gene and its protein in OSCC. However, non-genomic mechanisms (with some exceptions) could likely activate YAP. These different dependencies may have clinical consequences, as the data suggests that OSCC cells specifically dependent on WWTR1 appear to be associated with a distinct transcriptome and immune signature associated with response to immunotherapy. Given the considerable interest in therapeutically targeting immune pathways, these findings could provide a valuable response biomarker, as indicating that two Hippo pathway endpoints that are often grouped together have quite distinct roles. The results also could guide the use of therapies under development for targeting YAP1 or WWTR1/TAZ. For example, another of the important conclusions of this study is that TAZ-dependent cell lines may represent fiduciaries of their original tumors, which could be primary candidates for checkpoint inhibitors treatment via modern immunotherapy.

There are some points that should be addressed by the authors to solidify the conclusions and improve the study.

Essential revisions:

(1) The authors emphasize that a rationale for their study is the use of cell materials developed from Asian populations with different risk factors, such as use of betel. They hypothesize that these features of the OSCC cell set may result in different dependency profiles than those already published for OSCC from the DepMap project (Behan et al. 2019, Meyers et al., 2017). However, they do not explicitly compare results with those from the DepMap project. This analysis should be added, to determine if the hypothesized difference exists.

(2) A lot of the analysis pathway appears to be derivative of approaches from Behan et al., 2019. This includes data exclusion for core fitness genes, assignment of tractability groups, and other points. The authors need to make clear the novelty of the current work.

(3) From data shown in Figure 1C, most of the cell lines have a limited number of non-core fitness genes (40 or less), whereas 4 lines have a very large number of dependencies. What genomic features characterize those with large numbers of dependencies? If these are excluded, how does this affect the statistical analysis present in Figure 2? Similarly, 10 of the 21 cell lines appeared to have <10 core dependencies. Do these models specifically have unique genomic features, and YAP1 or WWTR1 dependency?

(4) In Figure 3—figure supplement 3, the authors note correlation between WWTR1 gene essentiality and expression across 273 cell models. What are the cancer types that are most dependent on WWTR1 (the point being, how specific is the observation of dependency to OSCC, versus being a general feature of squamous cancers)?

(5) In Figure 6, the authors analyze the YAP1 and WWTR1 gene signatures across a large group of OSCCs in the TCGA. Earlier in the study, the authors note that WWTR1 is often amplified as part of an amplicon including PIK3CA, TP63, and SOX2. To what extent is the transcriptional signature associated with amplification of these other genes, and dependent on their transcriptional activity, rather than that of WWTR1? Using their cell line models, they should complement TCGA analysis by using shRNA or drug inhibition to deplete WWTR1 versus other key genes in the amplicon, to determine which causes a specific loss of the immune-associated transcriptional signature. This is a really important data point for interpretation of the results.

(6) Besides papers cited by the authors, work relevant to the topic of this study includes Omori et al., 2020, for YAP1 as a driver, Saloura et al., 2019, showing YAP1 mutation is associated with a low CD8^+^ T cell inflamed phenotype, Martin et al., 2018, dissecting the Hippo pathway and demonstrating the targetability of YAP1 in head and neck cancer in a detailed Nat Comm paper, and an extensive review of the pathway at the end of 2019 by Santos-de-Frutos and colleagues. This work should be cited and discussed by the authors. It is important that the Discussion addresses more about how the generated data relate to previously published reports. For example, the fact that YAP, TAZ, PI3K, TP63, and SOX2 are amplified in OSCC is curious as all these genes are part of the extended signaling network of the Hippo pathway. Please see the examples of relevant references and consider a short paragraph discussing this point. [Huang et al., 2017; Fan , Kim and Gumbiner, 2013. And Li et al., 2019] Also when the role of TA in eliciting the immune response is mentioned, it would be important to reference one of the first publications (as far as the reviewer is aware of) that indicated the role of TAZ (and YAP, perhaps) via LATS1 KO in mice. The Hippo Pathway Kinases LATS1/2 Suppress Cancer Immunity. Moroishi T, (many authors) Guan KL. Cell. 2016 Dec 1;167(6):1525-1539.

(7) The authors should specifically state how many of their cell lines were highly dependent on WWTR1, how many on YAP1, and how many on both, out of the group of 21.

(8) The cell lines used in the screen are critical for the study. Therefore, it is suggested to provide in the Materials and methods and/or in the Results more information about how these lines were derived. It would help the readers a lot without referring to a previous publication (Fadlullah et al., 2016) and various commercial data for cell repositories. Please address if these cell lines were derived spontaneously (ala HeLa cells) or were engineered by transfections of immortalizing genes (ala HEK293 cell line, for example). If the latter was the case, please comment if the process of immortalization did not affect the actual transcriptional profile of the lines.

(9) As stated above, one of the critical conclusions this study is that TAZ-dependent OSCC lines may represent fiduciary (direct references) of their original tumors, and therefore patients with tumors driven by TAZ could be primary candidates for checkpoint inhibitors treatment via immunotherapy. Please address directly in the Discussion the fact that most of the tumors are quite heterogeneous in terms of oncogenic drivers and either the tumor biopsy or the cell lines derived from the tumor might not represent the "main" oncogenic drivers of the entire tumor.

---

## [Author Response]

Essential revisions:(1) The authors emphasize that a rationale for their study is the use of cell materials developed from Asian populations with different risk factors, such as use of betel. They hypothesize that these features of the OSCC cell set may result in different dependency profiles than those already published for OSCC from the DepMap project (Behan et al., 2019, Meyers et al., 2017). However, they do not explicitly compare results with those from the DepMap project. This analysis should be added, to determine if the hypothesized difference exists.

We thank the reviewers for the suggestion. We have added the analysis of enriched dependencies seen among betel quid-associated OSCC in the revised manuscript and discussed the results accordingly. In particular, by comparing the gene dependencies identified from betel quid-associated OSCC lines with those non-betel quid associated OSCCs, we found that the NF-κB pathway is significantly enriched in the betel-quid associated lines. When cross checking the fitness genes in this pathway with those from the DepMap project, where models not known to be associated with betel quid were used, all except one were not identified as fitness genes (CSNK2A1 being marginally depleted in one of the cell lines included in DepMap; SCC-4). Notably, the literature that reported on betel quid as a risk factor in OSCC have demonstrated that aberrant NF-κB pathway activation is common in these cancers. Furthermore, extracts from betel quid have been shown to directly activate NF-κB signaling in OSCC as reported in previous studies. By contrast, this is rarely reported among the OSCC studies from the Western countries, where betel quid is uncommon.

We have added the results of this analysis to Figure 2 (Figure 2D), and explained this in the Results section (subsection “Identification of unique dependencies in betel-quid associated OSCC”). We also discussed our findings in the Discussion section (third paragraph). The description of the analysis has been added to the Materials and methods section of the revised manuscript (subsection “Differential fitness genes analysis for betel quid associated OSCCs”). Figure legends and figure source data have also been included accordingly.

2) A lot of the analysis pathway appears to be derivative of approaches from Behan et al., 2019. This includes data exclusion for core fitness genes, assignment of tractability groups, and other points. The authors need to make clear the novelty of the current work.

The publication from Behan et al. is one of the most comprehensive essential screens that was conducted across many types of cancers (pan-cancer) and in their project, they have developed comprehensive and robust methods in filtering and analyzing data from such screens particularly in identifying tractable genes for further therapeutic development. As one of our aims in conducting the essential screens for OSCC was to identify gene targets that could be translated to clinical benefit through the development or identification of new therapies, we applied similar filtering criteria to our data including removing the core-fitness genes (as described by Hart et al. Cell. 2015 Dec 3;163(6):1515-26. doi: 10.1016/j.cell.2015.11.015), and assigning tractability groups as described by Behan et al. Whilst we may have adopted a similar approach, our work is novel in that (i) our study focuses on OSCC and it would be the first to describe non-core fitness genes for OSCC and it is also the largest dataset of dependencies in OSCC models; (ii) we included models derived from Asians (particularly those with betel-quid chewing habit) where the disease is most prevalent and describe possible distinctiveness in the non-core fitness genes in models that are representative of OSCC in Asians; (iii) we looked specifically at known oncogenes and recurrently amplified regions to interrogate the function of OSCC cancer genes, and (iv) we report distinct dependencies on YAP1 and WWTR1 among the different subsets of OSCC where these dependencies could have implications in cancer therapies including immunotherapy.

3) From data shown in Figure 1C, most of the cell lines have a limited number of non-core fitness genes (40 or less), whereas 4 lines have a very large number of dependencies. What genomic features characterize those with large numbers of dependencies? If these are excluded, how does this affect the statistical analysis present in Figure 2? Similarly, 10 of the 21 cell lines appeared to have <10 core dependencies. Do these models specifically have unique genomic features, and YAP1 or WWTR1 dependency?

We would like to clarify that the y-axis on Figure 1C refers to the number of dependent cell lines; while x-axis refers to the number of non-core fitness genes. For example, there are 366 genes that are essential in one out of 21 cell line (bottom most bar) – indicating fitness genes unique to a single cell line. In other words, we found 366 unique genes that is essential in only one cell line. On the other hand, only one gene (*NELFCD*) is a fitness gene for all the 21 cell lines (top most bar). This demonstrates the heterogeneity of the essentiality across the cell lines. To make this clearer, we have changed the y-axis label to “Number of dependent cell lines” and added more description in the figure legend.

4) In Figure 3—figure supplement 3, the authors note correlation between WWTR1 gene essentiality and expression across 273 cell models. What are the cancer types that are most dependent on WWTR1 (the point being, how specific is the observation of dependency to OSCC, versus being a general feature of squamous cancers)?

We observed the correlation of WWTR1 gene dependency with its gene expression in several cancer types including squamous cell lung carcinoma, breast carcinoma, glioblastoma, ovarian carcinoma and low-grade glioma in addition to OSCC. Therefore, this observation is not entirely unique to OSCC or squamous cell carcinoma. We have added this information on other cancers in Figure 3—figure supplement 3D and included the results in the fourth paragraph of the subsection “Differential dependency pattern on YAP1 and WWTR1”.

5) In Figure 6, the authors analyze the YAP1 and WWTR1 gene signatures across a large group of OSCCs in the TCGA. Earlier in the study, the authors note that WWTR1 is often amplified as part of an amplicon including PIK3CA, TP63, and SOX2. To what extent is the transcriptional signature associated with amplification of these other genes, and dependent on their transcriptional activity, rather than that of WWTR1? Using their cell line models, they should complement TCGA analysis by using shRNA or drug inhibition to deplete WWTR1 versus other key genes in the amplicon, to determine which causes a specific loss of the immune-associated transcriptional signature. This is a really important data point for interpretation of the results.

We thank the reviewer for the suggestion. We agree that it would be important to show that the correlation between WWTR1-dependency gene signatures with the immune signatures are indeed specific to WWTR1, and not due to other co-amplified genes such as PIK3CA, TP63 and SOX2. We did not previously include this analysis as using cell line model, most of the immune-associated transcriptional signature might not be directly measurable. Nevertheless, we have now taken two approaches to address this question as best as we could. First, we used a WWTR1-dependent model, ORL-214 to knockout WWTR1, PIK3CA, TP63 and SOX2 using two individual sgRNAs for each gene, and examined the down-regulation effect on PD-L1 gene expression. As we anticipated, only WWTR1-knock-out (KO) cells showed significant down-regulation of PD-L1 expression, but not in PIK3CA-, TP63-, nor SOX2-KO cells. These results are now reported in the subsection “ OSCC with WWTR1 dependency signature and immune biomarkers” and Figure 6—figure supplement 3A.

In addition, we also examined the correlation between the expression of PIK3CA, TP63 and SOX2 with PD-L1 expression, using an OSCC microarray dataset from Hiemer et al., 2015. Consistently, only WWTR1 gene expression showed significant correlation with PD-L1 gene expression. This correlation was not observed between PIK3CA, TP63, or SOX2 with PD-L1. These results are now reported in the aforementioned subsection and Figure 6—figure supplement 3B.

6) Besides papers cited by the authors, work relevant to the topic of this study includes Omori et al., 2020, for YAP1 as a driver, Saloura et al., 2019, showing YAP1 mutation is associated with a low CD8^+^ T cell inflamed phenotype, Martin et al., 2018, dissecting the Hippo pathway and demonstrating the targetability of YAP1 in head and neck cancer in a detailed Nat Comm paper, and an extensive review of the pathway at the end of 2019 by Santos-de-Frutos and colleagues. This work should be cited and discussed by the authors. It is important that the Discussion addresses more about how the generated data relate to previously published reports. For example, the fact that YAP, TAZ, PI3K, TP63, and SOX2 are amplified in OSCC is curious as all these genes are part of the extended signaling network of the Hippo pathway. Please see the examples of relevant references and consider a short paragraph discussing this point. [Huang et al., 2017; Fan, Kim and Gumbiner, 2013. And Li et al., 2019] Also when the role of TA in eliciting the immune response is mentioned, it would be important to reference one of the first publications (as far as the reviewer is aware of) that indicated the role of TAZ (and YAP, perhaps) via LATS1 KO in mice. The Hippo Pathway Kinases LATS1/2 Suppress Cancer Immunity. Moroishi T, (many authors) Guan KL. Cell. 2016 Dec 1;167(6):1525-1539.

We thank the reviewer for the suggestion to discuss our findings in the context of the highlighted literature. We have now elaborated on the relatedness of our findings with these literature and proposed opportunities for further studies that will address current knowledge gaps. These are now included in the Discussion of the revised manuscript. In addition to the suggested literature, we have also discussed and cited two additional papers that are relevant including Saladi et al., 2017, and Garcia-Escudero et al., 2018.

7) The authors should specifically state how many of their cell lines were highly dependent on WWTR1, how many on YAP1, and how many on both, out of the group of 21.

We thank the reviewer for this suggestion. Seven of the lines are highly dependent on YAP1, four are highly dependent on WWTR1, two are dependent on both YAP1 and WWTR1, while the rest did not show dependency on either YAP1 or WWTR1. This information is depicted in the bottom color-coded bar of the Figure 3A, where the dependency of the cell line on YAP1, WWTR1 and PIK3CA were determined from the extent of the genes depletion in the CRISPR screen, at a false-discovery rate of 5%, as defined using MAGECK analysis pipeline. We have now added this information in the second paragraph of the subsection “Differential dependency pattern on YAP1 and WWTR1”. We have also revised the figure legend to add clarity.

8) The cell lines used in the screen are critical for the study. Therefore, it is suggested to provide in the Materials and methods and/or in the Results more information about how these lines were derived. It would help the readers a lot without referring to a previous publication (Fadlullah et al., 2016) and various commercial data for cell repositories. Please address if these cell lines were derived spontaneously (ala HeLa cells) or were engineered by transfections of immortalizing genes (ala HEK293 cell line, for example). If the latter was the case, please comment if the process of immortalization did not affect the actual transcriptional profile of the lines.

As requested by the reviewer, we have now included information on how the lines were derived under the heading “Cell lines” in the Materials and methods section.

9) As stated above, one of the critical conclusions this study is that TAZ-dependent OSCC lines may represent fiduciary (direct references) of their original tumors, and therefore patients with tumors driven by TAZ could be primary candidates for checkpoint inhibitors treatment via immunotherapy. Please address directly in the Discussion the fact that most of the tumors are quite heterogeneous in terms of oncogenic drivers and either the tumor biopsy or the cell lines derived from the tumor might not represent the "main" oncogenic drivers of the entire tumor.

We agree with the reviewer and understand the importance of highlighting the assumption and limitation in our claims. We have now addressed in the Discussion section (seventh paragraph).